


# Reanalysis intercomparisons of stratospheric polar processing diagnostics

Zachary D Lawrence[1,2], Gloria L Manney[2,1], and Krzysztof Wargan[3,4]

[1]New Mexico Institute of Mining and Technology, Socorro, NM USA
[2]NorthWest Research Associates, Socorro, NM USA
[3]NASA/Goddard Space Flight Center, Greenbelt, MD USA
[4]Science Systems and Applications Inc., Lanham, MD, USA

*Correspondence to:* zachary.lawrence@student.nmt.edu

**Abstract.**

We compare herein polar processing diagnostics derived from the five most recent full-input reanalysis datasets: the National Centers for Environmental Prediction Climate Forecast System Reanalysis (CFSR), the European Centre for Medium-Range Weather Forecasts Interim Reanalysis (ERA-Interim), the Japanese Meteorological Agency's Japanese 55-year Reanalysis (JRA-55), and the National Aeronautics and Space Administration's Modern Era Retrospective-analysis for Research and Applications version 1 (MERRA) and version 2 (MERRA-2). We focus on diagnostics based on temperatures and potential vorticity (PV) in the lower to middle stratosphere that are related to formation of polar stratospheric clouds (PSCs), chlorine activation, and the strength, size, and longevity of the stratospheric polar vortex. Polar minimum temperatures ($T_{min}$) and the area of regions having temperatures below PSC formation thresholds ($A_{PSC}$) show large persistent differences between the reanalyses, especially in the southern hemisphere (SH), for years prior to 1999. Average absolute differences between the reanalyses in $T_{min}$ are as large as 6 K at some levels in the SH (2 K in the NH), and absolute differences in $A_{PSC}$ larger than 2% of a hemisphere (1% of a hemisphere in the NH). After 1999, there is a dramatic convergence toward better agreement between the reanalyses in both hemispheres throughout the lower stratosphere, with average $T_{min}$ differences generally less than 1 K in both hemispheres, and average $A_{PSC}$ differences less than 0.5% of a hemisphere. The comparisons of diagnostics based on isentropic PV for assessing polar vortex characteristics, including maximum PV gradients (MPVG) and the area of the vortex in sunlight (or sunlit vortex area, SVA), show more complex behavior: Some reanalyses show convergence toward better agreement in vortex diagnostics after 1999, while others show some persistent differences across all years. While the average differences are generally small for these vortex diagnostics, understanding such differences among the reanalyses is complicated by the need to use different methods to obtain vertically-resolved PV for the different reanalyses. We also evaluated other winter season summary diagnostics, including the number of days below PSC thresholds integrated over vertical levels, the winter mean volume of air below PSC thresholds, and vortex decay dates. For these summary diagnostics, the reanalyses generally agree best in the SH, where relatively small interannual variability has led to many winter seasons with similar polar processing potential and duration, and thus low sensitivity to differences in meteorological conditions among the reanalyses. In contrast, the large interannual variability of NH winters has given rise to many seasons with marginal conditions and high sensitivity to reanalysis differences. Our results indicate that the transition from the reanalyses assimilating Tiros Operational



Vertical Sounder (TOVS) data to Advanced TOVS and other data around 1998 – 2000 data had a profound effect on the agreement of the temperature diagnostics presented, and to a lesser extent the agreement of the vortex diagnostics. Our results lead to several recommendations for usage of reanalyses in polar processing studies, particularly related to the sensitivity to changes in data inputs and assimilation. Because of these sensitivities, we urge great caution for studies aiming to assess trends derived from reanalysis temperatures. We also argue that one of the best ways to present the sensitivity of scientific results on polar processing is to use multiple reanalysis datasets.

## 1 Introduction

Past, present, and future polar lower-stratospheric ozone depletion is a subject of critical scientific and human interest. Not only does chemical ozone depletion depend critically on temperatures and polar vortex dynamics in the lower stratosphere, but changes in lower stratospheric ozone also feed back and alter dynamical conditions in both the stratosphere and troposphere, which can significantly affect surface climate (Polvani et al., 2011; Albers and Nathan, 2013; WMO, 2014; Waugh et al., 2015, and references therein). Moreover, ozone depletion is affected by changing tropospheric and stratospheric temperatures, and in turn alters those temperatures via radiative forcing (e.g., Lacis et al., 1990; Forster and Shine, 1997; Levine et al., 2007; Hegglin et al., 2009; Telford et al., 2009; Riese et al., 2012; WMO, 2014). The southern hemisphere (SH) springtime polar vortex breakup disperses ozone-depleted air over populated regions, increasing surface UV exposure (e.g., Ajtić et al., 2003, 2004; Pazmino et al., 2005; WMO, 2007). Our ability to quantify chemical ozone loss in observations and to fully understand the mechanisms resulting in that destruction is a key to improving our modeling capability, which in turn will allow accurate forecasting of future ozone changes and their feedbacks on weather and climate. That ability depends critically on accurate knowledge of temperatures in and the dynamics of the lower stratospheric winter and springtime polar vortices. Even in the Antarctic, interannual variations in chemical ozone loss are controlled largely by variations in polar vortex dynamical conditions; thus detection of ozone recovery also requires accurate knowledge of variability and long-term changes in polar vortex dynamics and temperatures (e.g., Newman et al., 2004; Huck et al., 2005; WMO, 2014).

The chemistry leading to ozone loss involves the conversion of chlorine into forms that destroy ozone on the surfaces of cold aerosol particles and/or polar stratospheric clouds (PSCs) (see, e.g., Solomon, 1999; WMO, 2014, for reviews). These processes only occur at temperatures below a threshold that is dependent on pressure and on water vapor ($H_2O$) and nitric acid ($HNO_3$) concentrations (e.g., Hanson and Mauersberger, 1988; Solomon, 1999). Furthermore, these processes can only result in widespread and persistent chlorine activation when/where the cold air is confined so that mixing with outside air cannot dilute the activated fields – that is, inside the "containment vessel" of the winter and springtime stratospheric polar vortex (e.g., Schoeberl and Hartmann, 1991; Schoeberl et al., 1992). Finally, the reactions by which active chlorine destroys ozone require sunlight. The formation and maintenance of the dynamical and chemical environment described above is referred to as "polar processing" since all of these conditions are required for chemical ozone depletion to take place in the lower stratosphere. Since reanalyses from data assimilation systems (DAS) are the best available tools for modeling and understanding stratospheric dynamics, as well as for driving models of past and present ozone loss, the representation of lower stratospheric temperatures



and vortex dynamics in these reanalyses is critical to furthering our understanding of and ability to predict ozone depletion and eventual ozone recovery.

Over approximately the past two decades, numerous studies have compared meteorological products from DAS in the polar stratosphere (e.g., Manney et al., 1996; Pawson et al., 1999; Davies et al., 2003; Feng et al., 2005), and/or compared such

products with observations (e.g., Knudsen et al., 1996, 2001, 2002; Gobiet et al., 2007; Boccara et al., 2008; Tomikawa et al., 2015); see, e.g., Lawrence et al. (2015) and Lambert and Santee (2017) for further review.

One important finding of earlier studies was that the NCEP/NCAR (National Centers for Environmental Prediction / National Center for Atmospheric Research) and NCEP/Department of Energy reanalyses are unsuitable for polar processing studies because of their poor representation of the stratosphere (very low model top and few model levels) and outdated assimilation

approaches (e.g., assimilation of retrieved temperature from operational sounders) (e.g., Manney et al., 2003, 2005a, b). The European Centre for Medium-range Weather Forecast's (ECMWF's) 40-year Reanalysis (ERA-40) reanalysis was also shown to be unsuitable for such studies, partly because of unrealistic oscillations in the temperature profiles (e.g., Manney et al., 2005a, b; Feng et al., 2005; Simmons et al., 2005). In the past few years, some studies have begun focusing on the latest generation of reanalyses, which have vast improvements in models and assimilation methods and more comprehensive data

inputs (for a review of reanalysis characteristics, see Fujiwara et al., 2017). WMO (2014) showed comparisons of potential PSC volume and of springtime vortex breakup dates (calculated as in Nash et al., 1996) between NCEP/NCAR and two modern reanalyses, ECMWF's Interim reanalysis (ERA-Interim) and the NASA Global Modeling and Assimilation Office's (GMAO's) Modern Era Retrospective analysis for Research and Applications (MERRA); NCEP/NCAR was shown to give much lower PSC volumes than in the more modern reanalyses, and the vortex breakup dates differed substantially among

each of the reanalyses. Lawrence et al. (2015) compared a large suite of diagnostics, based on polar vortex characteristics and temperatures, that are important for polar processing between MERRA and ERA-Interim (hereinafter referred to as ERA-I) for the then-available 34 years of those reanalyses. These comparisons showed significant changes in agreement between the reanalyses over that period, with overall good agreement in the period since 2002, when the amount of data ingested into the two reanalyses' DAS was much greater; the largest improvements in agreement were particularly seen in Antarctic temperature

diagnostics. In a paper describing global temperature and wind comparisons as part of the Stratosphere-troposphere Processes And their Role in Climate (SPARC) Reanalysis Intercomparison Project (S-RIP) (Fujiwara et al., 2017), Long et al. (2017) also emphasized changes in agreement between reanalyses related to data input changes, especially improvements in temperature agreement after the transition from TOVS to ATOVS around 1998 to 2000; they also pointed out issues with discontinuities in some reanalyses that were run in multiple streams. Changes such as these noted by Lawrence et al. (2015) and Long et al.

(2017) argue for great hesitancy in using temperatures and other fields from individual reanalyses for diagnosing long-term changes and trends.

The ability to compare polar processing diagnostics with observations is very limited for several reasons. Somewhat paradoxically, the vast improvements in DAS usage of available observations have resulted in there being very few truly independent temperature datasets. Furthermore, many of the datasets that are available, even those ingested into the DAS, generally suffer

from very limited spatial and/or temporal coverage (e.g., balloon-borne and lidar measurements) and/or issues with resolution,




precision, and length of data records (e.g., limb-sounding research satellites). Nevertheless, several recent studies have compared some of the latest generation reanalyses with observations: For example, Hoffmann et al. (2017) compared MERRA, MERRA-2 (the recent successor to MERRA), ERA-Interim, and NCEP/NCAR reanalyses with temperatures and winds from long-duration Concordiasi ballon flights in the Antarctic lower stratopshere from September 2010 through January 2011; un-

surprisingly, they found much larger temperature biases for NCEP/NCAR than in the other reanalyses, not only because that reanalysis is unsuitable for stratospheric studies, but also because the other reanalyses they considered assimilated Concordiasi measurements. Lambert and Santee (2017) compared MERRA, MERRA-2, ERA-Interim, JRA-55 (the Japan Meteorological Agency's latest reanalysis assimilating both surface and upper air observations, hereinafter referred to as a "full-input" reanalysis), and CFSR (referring collectively to NCEP's Climate Forecast System Reanalysis and Climate Forecast System Version

2) with COSMIC (Constellation Observing System for Meteorology, Ionosphere and Climate) GPS-RO (Global Positioning System–Radio Occultation) temperatures, and presented an innovative analysis using thermodynamic calculations to derive an independent temperature reference from satellite observations of $HNO_3$, $H_2O$ (from the Aura Microwave Limb Sounder, MLS), and PSC aerosols (from Cloud-Aerosol Lidar with Orthogonal Polarization). They found temperature biases in the reanalyses with respect to COSMIC of -0.6 to +0.5 K, and biases ranging from -1.6 to +0.1 K with respect to the derived temperature

references for two PSC types.

The use of multiple data sources and novel methods allowed Lambert and Santee (2017) to compare temperatures over a wide range of winter polar vortex conditions in both hemispheres for 2008 through 2013. Studies comparing with other data sources, such as long-duration balloon flights (Hoffmann et al., 2017, and references therein), are generally restricted to more limited spatial and temporal regimes. In addition, many of the latest generation reanalyses assimilate data sources such as

COSMIC GPS-RO and long-duration balloon flights (Fujiwara et al., 2017; Hoffmann et al., 2017; Lambert and Santee, 2017), thus complicating interpretation of differences from those data sources. Also, as noted by Lawrence et al. (2015), some of the most useful diagnostics of polar processing, while conceptually simple, depend on having full and dense coverage of the polar regions (e.g., minimum high-latitude temperatures or area of temperatures below a PSC threshold), and/or are based on vortex diagnostics that are defined by potential vorticity (PV) (e.g., vortex area or vortex-edge PV gradients) that do not have

corresponding observations. Furthermore, reanalyses are used in polar processing studies that span the 35 (or more) years of their duration, during much of which there are no data available with widespread coverage for comparison. Because of these limitations, comparisons of reanalyses remain one of our most valuable tools for assessing their representation of the dynamical conditions that control polar chemical processing and ozone loss.

Since the work described in Lawrence et al. (2015), the MERRA-2 reanalysis has become available and widely used, in-

cluding for polar processing and polar vortex studies (e.g., Manney and Lawrence, 2016; Lawrence and Manney, 2017). Since that work was completed, the CFSR reanalysis was made available on the native model levels, and we have also obtained and processed the complete JRA-55 record. While Long et al. (2017) compared temperatures in all of the latest generation reanalyses, they focused on zonal means and the whole stratosphere rather than on the polar lower stratosphere and diagnostics specifically relevant to polar processing. To our knowledge, no studies have been done that compared lower stratospheric polar

processing diagnostics in all five recent full input reanalyses (MERRA, MERRA-2, ERA-I, JRA-55, and CFSR).



In this paper, we compute and analyze the diagnostics used by Lawrence et al. (2015) to provide a more complete and quantitative characterization of reanalysis differences during the satellite era. In addition to including the MERRA-2, JRA-55, and CFSR reanalyses, the calculation and analysis of diagnostics has been updated to include sensitivity tests (e.g., to temperature and vortex edge thresholds) and to include assessment of the variability in reanalysis differences and the statistical

significance of those differences. Section 2.1 briefly describes the reanalysis datasets and the assimilation system inputs most relevant to assessment of polar processing diagnostics. Section 2.2 describes the diagnostics we calculate and the methods used to analyze them. Our results are presented in Section 3, comprising temperature (Section 3.1), vortex (Section 3.2) and derived (Section 3.3) diagnostics. Section 4 gives a summary and conclusions.

## 2   Data and Methods

### 2.1   Reanalysis Datasets

Fujiwara et al. (2017) provide detailed descriptions of the models, assimilation systems, and data inputs for the reanalyses used here in their overview paper on S-RIP. We compare the five most recent high-resolution "full-input"reanalysis climatologies for all winters with data available since the beginning of the "satellite era" in 1979. All analyses are done using daily 12-UT fields from each reanalysis dataset. Because of the importance of resolution, especially the vertical grid, in representing the

polar lower stratosphere and threshold processes in general (see, e.g., Manney et al., 2017), except where unavailable (e.g., PV for MERRA), we start our analyses from reanalysis data on the native model levels and at or (in the case of spectral models) near the native horizontal resolution.

#### 2.1.1   MERRA and MERRA-2

The National Aeronautics and Space Administration (NASA) GMAO's MERRA (Rienecker et al., 2011) dataset is a global

reanalysis covering 1979 through 2015. It is based on the GEOS (Goddard Earth Observing System) version 5.2.0 assimilation system, which uses 3D-Var assimilation with Incremental Analysis Update (IAU) (Bloom et al., 1996) to constrain the analyses. The model uses a $0.5° \times 0.667°$ latitude/longitude grid with 72 hybrid sigma-pressure levels, with about $\sim 1.2\,\mathrm{km}$ grid spacing in the lower stratosphere. PV data from MERRA were not archived on the model levels and grid, and thus we use the PV provided on a $1.25° \times 1.25°$ latitude/longitude grid on 42 pressure levels.

MERRA-2 (Gelaro et al., 2017) uses a similar model and assimilation system to MERRA, with updates also described by Bosilovich et al. (2015), Molod et al. (2015), and Takacs et al. (2016).Changes between MERRA and MERRA-2 that may significantly affect representation of the polar lower stratosphere include addition of new observation types in MERRA-2 (see Section 2.1.5, Figure 1 and Fujiwara et al. (2017)); a different treatment of conventional temperature data in MERRA-2; and assimilation of data in MERRA-2 using upgraded background error statistics, which control the magnitude and spatial extent

of the impact of observations on the assimilated product. In addition, the effects of using a more uniform horizontal grid in MERRA-2 may be important. The impact of using the "cubed-sphere" grid in MERRA-2 is particularly seen in improved



PV fields near the poles, especially in situations (common in the winter stratosphere) with strong cross-polar flow (e.g., see Figure 20 of Gelaro et al., 2017). Different radiative transfer models are used for the assimilation of Stratospheric Sounding Unit (SSU) data in MERRA and MERRA-2. MERRA used GLATOVS (Goddard Laboratory for Atmospheres TOVS, Susskind et al., 1983) for SSU assimilation, while MERRA-2 uses the CRTM (Community Radiative Transfer Model, Han et al., 2006;

Chen et al., 2008) for SSU and for all other radiance data. In particular, the CRTM takes into account long-term changes in the height of the SSU's weighting functions resulting from gradual reduction of pressure in the instruments' $CO_2$ cells (as described by, e.g., Kobayashi et al., 2009) that GLATOVS did not. For the other radiances, MERRA also used CRTM, but a much older version than that used in MERRA-2. Gelaro et al. (2017) and references therein give details of these and other changes.

The MERRA-2 data products are described by Bosilovich et al. (2016). All MERRA-2 data products used here are on model levels (the same vertical grid as for MERRA) and a $0.5° \times 0.625°$ latitude/longitude grid. Data from MERRA-2 from its spin-up year, 1979, are not in the public MERRA-2 record, but we use data from late 1979 to start the analysis with the NH 1979/1980 winter. We use the MERRA-2 "Assimilated" (ASM) data collection (Global Modeling and Assimilation Office (GMAO), 2015) here, as recommended by GMAO, particularly for studies that require consistency between mass and

wind fields (see, e.g., https://gmao.gsfc.nasa.gov/reanalysis/MERRA-2/docs/ANAvsASM.pdf and Fujiwara et al., 2017). For MERRA, however, the ASM fields are not available on the model grid, but only at degraded horizontal and vertical resolution; we thus use the MERRA ANA collection for most variables here (see also Section 2.2.1). Differences between ANA and ASM fields are small, but can be non-negligible (e.g., Manney et al., 2017).

### 2.1.2 CFSR

NCEP-CFSR/CFSv2 (hereinafter CFSR) (Saha et al., 2010) is a global reanalysis covering the period from 1979 to the present. The data are produced using a coupled ocean-atmosphere model and 3D-Var assimilation. CRTM is used for satellite radiance assimilation. The model resolution is T382L64; the data used here are on a $0.5° \times 0.5°$ horizontal grid on the model levels (available through 2015); vertical grid spacing in the lower stratosphere ranges from about 0.8 km near 100 hPa to 1.3 km near 10 hPa. CFSR did make an undocumented update to their assimilation scheme in 2010 (Long et al., 2017). Furthermore, in the

transition from CFSR to CFSv2 in 2011, the resolution, forecast model, and assimilation scheme were all upgraded; CFSv2 is, however, intended as a continuation of CFSR and can be treated as such for most purposes (Saha et al., 2014; Fujiwara et al., 2017; Long et al., 2017).

### 2.1.3 ERA-Interim

ERA-Interim (see Dee et al., 2011) is another global reanalysis that covers the period from 1979 to the present. The data

are produced using 4D-Var assimilation with a T255L60 spectral model. ERA-I uses the RTTOV (Radiative Transfer for TOVS) radiative transfer model, version 7 for radiance assimilation. Here we use ERA-Interim data on a $0.75° \times 0.75°$ latitude/longitude grid (near the resolution of the model's Gaussian grid) on the 60 model levels. The spacing of the model levels in the lower stratosphere is about 1.2 to 1.4 km.





### 2.1.4 JRA-55

JRA-55 (Ebita et al., 2011; Kobayashi et al., 2015) is a global reanalysis that covers the period from 1958 to the present, and is produced using a 4D-Var assimilation. The data from the JRA-55 T319L60 spectral model are provided on an approximately 0.56° Gaussian grid corresponding to that spectral resolution. JRA-55 uses RTTOV version 9.3 for satellite radiance assimilation. The JRA-55 fields on the model vertical levels have a vertical resolution of ∼1.2 to 1.4 km in the lower stratosphere.

### 2.1.5 Timeline of Satellite Data Inputs to DAS

Operational satellite observations are the primary data constraints on reanalyses at stratospheric levels. Figure 1 shows these satellite inputs (see Table 1 for a list of satellite acronyms) for each of the reanalyses used here, shown as stacked timelines to facilitate comparison of changes in data inputs between reanalyses. Up through about 1994, all of the reanalyses relied primarily on the TOVS instruments (SSU, MSU, VTPR&HIRS), and (excepting CFSR) SSM/I&SSMIS. Between 1995 and 2002, there are several changes in the data inputs, and the inputs begin to vary more among the reanalyses. MERRA and MERRA-2 have much the same inputs for most of the period, but for recent years MERRA-2 assimilates Meteosat, IASI, CrIS, ATMS, and GPS-RO observations. IASI, CrIS, and ATMS are not assimilated in any of the other reanalyses. A change with a large impact on stratospheric temperatures overall is the transition from TOVS (with MSU and SSU) to ATOVS (with AMSU-A and AMSU-B/MHS); Figure 1 shows that this transition is handled/timed differently among the reanalyses: For instance, although all reanalyses introduce AMSU-A at just about the same time, JRA-55 stops assimilating the TOVS instruments' data by 2000, whereas the others continue until 2006 (except for an immediate cut off of SSU by 1999 in CFSR). Long et al. (2017) showed that this transition had a profound impact on the differences in zonal mean temperatures among the reanalyses, with a shift toward much better agreement after the transition.

## 2.2 Methods

The methods and diagnostics used herein are largely the same as those used by Lawrence et al. (2015). In the subsections to follow, we describe the reanalysis fields we use, and how we prepare them and derive the polar processing diagnostics from them. We also provide some information on additional analysis techniques that we use to help interpret our results.

### 2.2.1 Preparation of Meteorological Fields

The two meteorological fields necessary to derive most of the diagnostics used herein are temperatures and potential vorticity (PV) on isentropic surfaces. In the results we present later, we show temperature diagnostics on pressure surfaces, and vortex diagnostics on isentropic surfaces; as will be discussed, we also calculate and use some temperature diagnostics on isentropic levels.

Of the 5 reanalyses described in Section 2.1, only MERRA-2 provides potential vorticity on their model levels. MERRA provides PV calculated within the assimilation system, but interpolated to 42 pressure levels on a reduced horizontal resolution grid, which we interpolate to the same grid (the native model grid) on which we use the MERRA temperatures. CFSR,




ERA-Interim and JRA-55 only provide PV on a sparse set of isentropic levels, with very few common levels between them. ERA-Interim provides absolute vorticity on model levels, and CFSR provides relative vorticity; thus to get the vertically-resolved PV fields that we need, we derive PV for these reanalyses using their provided vorticity, temperature, and pressure fields on the model levels. In the case of JRA-55, we use the zonal and meridional wind components to first calculate rela-

tive vorticity, which we then use in combination with temperature and pressure to calculate PV on the model levels. While a thorough evaluation of biases that may arise from using different types of PV calculations for different reanalyses would be valuable, it is beyond the scope of this paper. We think, however, that data users will most likely use the most direct calculation to get from the provided fields to the model-level PV (as we have), and thus the fields we are comparing are those that are most likely to be used in practice. When calculating polar processing diagnostics, we scale the PV fields into vorticity units as

in Dunkerton and Delisi (1986) by dividing the PV by a standard value of static stability calculated from assuming a vertical temperature gradient of $1\,\mathrm{K\,km^{-1}}$ and a pressure of $54\,\mathrm{hPa}$ on the 500 K isentropic level. We use this scaling so that the scaled PV (sPV) values are of the same order of magnitude at different levels throughout the stratosphere.

Since the reanalyses are all on different model levels, we use the reanalyses' temperature and pressure fields to vertically interpolate their temperature and PV fields to a common set of fixed pressure and isentropic surfaces. We use a

standard set of pressure and isentropic levels that have been used for several NASA satellite instrument datasets (see, e.g., https://cdn.earthdata.nasa.gov/conduit/upload/4849/ESDS-RFC-009.pdf) including the Aura Microwave Limb Sounder (Livesey et al., 2015); these are 12 levels per decade in pressure, and their climatologically corresponding isentropic surfaces. For polar processing diagnostics, we limit our focus to the thirteen levels between roughly $120 - 10\,\mathrm{hPa}$, or $390 - 850\,\mathrm{K}$.

### 2.2.2 Temperature and Vortex Diagnostics

The depletion of ozone in the lower stratosphere follows from a complex chain of processes that are highly dependent on meteorological conditions (see, e.g., Solomon, 1999; WMO, 2014, for reviews). The activation of chlorine requires the presence of polar stratospheric clouds (PSCs), which form when temperatures are sufficiently low, and grow when temperatures stay low for sufficiently long periods of time (Hanson and Mauersberger, 1988; Solomon, 1999; WMO, 2014, and references therein). The catalytic ozone destruction cycles involving both chlorine and bromine further require sunlight, which is usually provided

later in winter and spring when sunlight returns to the high latitude polar regions. These chemical processes also require isolation from lower latitude air, which is provided by the stratospheric polar vortex; the edge of the polar vortex acts as a barrier preventing transport and mixing, and thus the vortex acts as a containment vessel for the polar air where these processes take place (e.g., Schoeberl and Hartmann, 1991; Schoeberl et al., 1992). We thus examine polar processing diagnostics that primarily focus on lower stratospheric temperatures and the state of the polar vortex to assess the meteorological conditions

conducive to ozone depletion. Unless specified otherwise, we focus on the months from December through March (DJFM) for the Northern Hemisphere (NH), and May through October (MJJASO) for the Southern Hemisphere (SH); these time periods cover roughly the full period during which polar processing takes place for most polar winters (see Section 3.1).

The temperature diagnostics that we use include minimum temperatures ($T_{min}$) poleward of $\pm 40°$, and the areas of temperatures (poleward of $\pm 30°$) below PSC formation thresholds ($A_{PSC}$, or Area $T \leq T_{PSC}$). For $A_{PSC}$, we specifically use the





formation temperatures for solid nitric acid trihydrate (NAT, Hanson and Mauersberger, 1988) and ice particles on pressure levels, which we define using climatological profiles of $HNO_3$ and $H_2O$ mixing ratios. As a rule of thumb, the NAT threshold between 120 and 10 hPa ranges from roughly 198 – 187 K respectively, and the ice threshold tends to be between 6 – 8 K below the NAT threshold. We stress that these thresholds are approximations, but they are convenient as proxies for PSC for-

mation and chlorine activation. While most of the results we show herein for the temperature diagnostics are on pressure levels, we also calculate them on isentropic surfaces; for the diagnostics involving PSC thresholds, we assign the PSC thresholds on pressure levels to the isentropic surfaces that are roughly co-located (e.g., 520 K corresponds to 46.1 hPa). This is an additional approximation, but it allows us to keep the intercomparisons simple without having to calculate daily varying PSC thresholds for pressures/temperatures on isentropic levels, or pre-computing climatological PSC threshold values from the reanalyses'

fields. To mitigate issues with these approximations and test sensitivity to the thresholds, we also compute $A_{PSC}$ with $\pm 1$ K offsets to the PSC formation temperatures.

The vortex diagnostics that we use include maximum gradients in sPV as a function of equivalent latitude (maximum sPV gradients, or MPVG), which assess the strength of the vortex edge (e.g., Manney et al., 1994, 2011; Lawrence et al., 2015), and the area of the vortex exposed to sunlight (sunlit vortex area, or SVA). To calculate MPVG, we bin sPV as a function of

equivalent latitude (EqL, the latitude that would enclose the same area between it and the pole as a given PV contour; Butchart and Remsberg, 1986), numerically differentiate, and catalog the maximum value between $\pm 30$ and $\pm 80°$. We use $\pm 30°$ as a lower limit because relatively large PV gradients can be found in the tropics, which can dominate early/late in the season when the vortex is forming/decaying; we use $\pm 80°$ as an upper limit because the small areas represented by points poleward of $\pm 80°$ can vary much more dramatically than the lower EqLs, sometimes producing large gradients (e.g., Nash et al., 1996)

that are not indicative of the vortex edge. To calculate SVA, we calculate the area of the vortex that extends equatorward of the daily polar night latitude at 12:00UT. We use the vertical profile of vortex edge values defined by Lawrence and Manney (2017) rounded to the nearest $1.0 \times 10^{-5}$ s$^{-1}$ to specify the vortex edge in both the NH and SH (in the SH, the values are multiplied by -1). This vortex edge profile was defined from MERRA-2 data for the NH polar vortex, but we have confirmed it is appropriate for the SH vortex, whose edge varies (in a climatological sense) similarly with height.

**2.2.3   Analysis Techniques**

Rather than comparing the diagnostics derived from each of the reanalyses to an average across a subset of the reanalyses (for S-RIP, this is referred to as the "Reanalysis Ensemble Mean"), we opt for comparing CFSR, ERA-Interim, JRA-55, and MERRA directly with MERRA-2. We do this for a few reasons: first, 4 – 5 reanalyses is a relatively small sample that can make the ensemble average sensitive to outliers. The second and related reason we compare with MERRA-2 directly is that

comparing with an average across the reanalyses obscures the actual scale of differences among the individual datasets since the mean of the reanalyses will "centralize" the diagnostics. Finally, it is unclear to us whether it is appropriate to include both or only one of MERRA or MERRA-2 in the average; while MERRA-2 is an improvement over MERRA with numerous differences in the data ingested, they do use similar models, assimilation schemes, etc. Thus, for the results herein, we will primarily show climatologies and reanalysis differences from MERRA-2 (i.e., reanalysis minus MERRA-2). Henceforth, we





will colloquially refer to the group of CFSR, ERA-Interim, JRA-55, and MERRA as "the reanalyses", but this should not be confused as us making a value judgment of MERRA-2 as the "best" or "standard" reanalysis for other reanalyses to be compared against.

Part of our analysis uses a statistical significance test to determine whether the average differences between the reanalyses and MERRA-2 are statistically different from zero over a winter season. To accomplish this, we use a non-parametric bootstrap resampling technique that is useful for time series datasets called the stationary bootstrap (Politis and Romano, 1994). Bootstrapping methods for time series have generally relied on resampling blocks of consecutive observations to construct many artificial time series so that accuracy estimates can be made for sample statistics/estimators (e.g., Lahiri, 2003, and references therein). Rather than resampling random blocks (which may or may not overlap) of a fixed size to construct artificial time series, the stationary bootstrap constructs pseudo time series by resampling blocks of random size determined from a geometric distribution with specified mean. Herein, we bootstrap the time series of differences from the reanalyses and MERRA-2; we treat the differences from each reanalysis, on each vertical level, and each year individually. In all cases, we use the stationary bootstrap with a specified geometric distribution mean of 5 (i.e., the expected block length is 5 days), and resample all the time series of differences $10^5$ times. We then use the bootstrap percentile method to construct 99% confidence intervals (CIs) of the average differences; the percentile method is known to have issues in cases with small sample sizes, but since we use a more strict 99% CI and our time series are longer than 120 days, we expect our estimates are robust (see discussion in DiCiccio and Efron, 1996, and references therein). When these 99% CIs do not contain zero, we consider the average differences for the reanalysis minus MERRA-2 (for a specific level and year) to be indicative of persistent positive or negative differences.

## 3 Results

In the next two subsections, we show comparisons of temperature and vortex diagnostics as yearly time series of average differences and standard deviations calculated over the polar processing periods in each hemisphere (DJFM for the NH, MJJASO for the SH). We use these averages and standard deviations to evaluate the agreement between the reanalyses. To demonstrate what we mean by agreement, in Figure 2 we have plotted yearly time series of the average differences and standard deviations of differences in a theoretical diagnostic calculated by subtracting one comparison reanalysis from two theoretical reanalyses (cyan and magenta lines). The magenta reanalysis tends to have smaller average differences from the comparison reanalysis throughout the period than the cyan reanalysis, but the cyan reanalysis tends to have smaller standard deviations. Even when the magenta reanalysis has relatively small average differences between 1995 and 2000, it has large standard deviations, indicating that the spread around the average is quite large during this time. In contrast, for the same period, the cyan reanalysis has relatively small standard deviations, but large negative average differences, indicating that the diagnostic calculated from the cyan reanalysis tends to be systematically higher than the comparison reanalysis. Later on, the average differences and standard deviations for both reanalyses approach zero, indicating a convergence towards better agreement with the comparison reanalysis, but also better agreement between the cyan and magenta reanalyses themselves.





Since we examine averages and standard deviations of differences across multiple years and vertical levels, we illustrate in Figure 3 how we display the reanalysis differences in Sections 3.1 and 3.2. For each year, level, and reanalysis, the diagnostics calculated from MERRA-2 (minimum temperatures in this example; top panel) are subtracted from those calculated from the other reanalyses (ERA-Interim in the example; bottom left panel). The averages and standard deviations are then found over

the full polar processing period bounded by the vertical black lines) for each vertical level. These values are then plotted as individual pixels in a column (bottom right panel) summarizing the differences from a single year.

## 3.1 Temperature Diagnostics

Figure 4 shows the climatological values of minimum temperatures from MERRA-2. The well known difference in stratospheric temperatures between NH and SH (e.g., Andrews, 1989) is seen clearly, with the climatological period with tempera-

tures below the NAT PSC threshold spanning approximately December through mid-February in the NH and mid-May through early October in the SH. The lowest temperatures are centered near 20 hPa at about the time of the solstice in the NH, and near 25 hPa approximately a month after the solstice in the SH. NH winter temperatures are lowest earlier in the season because of the prevalence of sudden stratospheric warmings (SSWs) in January and February in that hemisphere.

Figure 5 shows the yearly time series of winter mean differences in minimum temperatures for the Antarctic, as "pixel" plots

constructed as described above. The most striking feature is an overall improvement in the agreement after 1998 to 1999, with some differences having magnitudes up to about 6 K in the earlier years, as opposed to near 1 K in later years. A corresponding decrease is seen in the standard deviations, with values up to over 3 K before 1999, and typically near 1 K thereafter. This time corresponds to the transition from assimilating TOVS to ATOVS radiances. In addition, in 2002, all of the reanalyses except JRA-55 began assimilating the hyperspectral AIRS radiances, vastly increasing the number of observations used in the

Antarctic.

During years from roughly 1993 through 1998, larger differences and standard deviations are seen above about 30 hPa in ERA-I, and to a lesser degree in the other reanalyses. While the main source of stratospheric information for all the reanalyses in this time period is the SSU and MSU instruments, there are several differences in how these data are assimilated: The different radiative transfer models used for satellite radiance assimilation handle inter-satellite drifts due to SSU $CO_2$ cell

pressure leaks differently; MERRA and MERRA-2 also handle these differently because of the change in radiative transfer model used. There are also differences in which SSU channels a bias correction is applied to (MERRA-2 does not bias correct Channel 3, but JRA-55 and ERA-I do). It is even more difficult to speculate about changes in CFSR, since it has multiple discontinuities and biases related to stitching together execution streams and applying a bias correction in a model with a warm bias (Long et al., 2017). Thus, while we cannot pin down particular changes that are associated with this increase in variance,

there are numerous differences that could contribute to this behavior.

The improvements after 1998 are largest at higher levels (where the differences and standard deviations are themselves largest), becoming less prominent, and less sudden, below about 50 hPa. Before 1999, CFSR, ERA-I, and JRA-55 all show negative differences between about 70 and 30 hPa sandwiched between positive differences above and below that. Investigations in progress (Long et al., in preparation) show that both MERRA-2 and ERA-I temperatures in the SH polar stratosphere



have oscillations of up to about 3 K, which are in opposite directions, leading to the layered structure of the differences seen here. The other reanalyses do not show vertical oscillations, but the oscillations in MERRA-2, of course, show up in the differences. (Note that the absence of oscillations in the other reanalyses does not imply better agreement with sondes, Long et al., in preparation).

Average differences between MERRA-2 and ERA-I, JRA-55, and MERRA are not statistically significant for many regions and years after 1999, especially between about 50 and 20 hPa. In comparison with the other reanalyses, CFSR continues to show larger standard deviations and more significant average differences from MERRA-2 after 1999. CFSR and JRA-55 show a change in sign of the differences in the upper levels (∼20–10 hPa), while ERA-I shows a similar change in sign, but only above about 15 hPa. After 1999, the MERRA − MERRA-2 differences also change signs from negative (positive) to positive
(negative) at pressures less (greater) than roughly 40 hPa; however, their differences during this time period are typically within 1 K. Despite substantial changes in the model and assimilation systems, MERRA shows much closer agreement with MERRA-2 before 1998 than do the other reanalyses, but still shows a sudden decrease in the differences at that time. This suggests that the improved resolution from the ATOVS instruments and the increase in the number of observations are major factors in the improved agreement among all reanalyses, but that the differences in the models and data handling (which are smaller between
MERRA and MERRA-2 than between the other reanalyses and MERRA-2) are also an important factor.

The year 1984 stands out, especially in JRA-55, as having much lower differences and larger standard deviations in the higher levels than during the surrounding years. Inspection of the daily differences for individual years (not shown) indicates a period in July, August, and September in that winter with negative differences instead of large positive ones in CFSR, JRA-55, and MERRA, and a period in June and July in ERA-I with much smaller positive differences than those in other years, at
the highest levels. While in 1983 and 1985, MERRA-2 assimilated radiances from two SSU instruments and several channels from those instruments, in 1984 MERRA-2 assimilated data from only one (NOAA-7) SSU instrument, and channel 2 (which peaks above, but has considerable influence below, 10hPa) on that instrument was off. Furthermore, there was a change in how MERRA-2 assimilated MSU (whose highest peaking channel samples the lower stratosphere) radiances at about this time. While not conclusive, these changes could have contributed to the anomaly in the MERRA-2 differences from the other
reanalyses in 1984.

Average differences in minimum temperatures in the NH (Figure 6 show more complicated patterns of changes over the years. The differences are much smaller throughout the 36-year period, with maximum differences near 2 K at the highest levels shown, mainly in the period from about 1994 to 2004, and more frequent times/regions throughout the period studied when the average differences are not significant. The standard deviations do decrease somewhat after around 1999, though
(as was the case in the SH) remain larger in CFSR than in the other reanalyses. There are indications of sudden changes around 1999, but they are less abrupt than those in the SH, and it is often less clear that there is a uniform trend towards better agreement. Prior to about 1999, ERA-I, JRA-55, and CFSR show a similar pattern to that in the SH of positive differences at the lowest and highest levels surrounding negative differences between those layers; CFSR and JRA-55 show more regions of negative than positive differences after 1999. ERA-I shows a decrease in the regions with significant average differences after
about 1999, but, while the patterns of such regions change for the other reanalyses, a clear decrease in them is not evident. As



in the SH, the differences between MERRA-2 and MERRA are smaller than those between MERRA-2 and the other reanalyses in the period before about 1998.

Figure 7 shows MERRA-2 climatological values of the area with temperatures below the NAT PSC threshold ($A_{NAT}$) for the NH and SH winter seasons. As expected, these echo the patterns of minimum temperatures seen in Figure 4, with the largest

values in the NH in early January and in the SH in middle to late July. The great variability in the NH (see the grey envelopes in the line plots) results in the largest values being well above the climatological average, about 7–8% of a hemisphere, but still much lower than the largest values in the SH of over 12% of a hemisphere.

Note that comparing differences in NH $A_{NAT}$ among the reanalyses is more difficult than doing so for the SH or for the other NH diagnostics. Because there is significant interannual variability in the onset, termination, and magnitudes of low temper-

atures in the NH (see both Figures 4 and 7), there are many NH winters with relatively few days having temperatures below $T_{NAT}$, and thus many days with NH $A_{NAT}$ being zero. Thus, comparing differences among the reanalyses for the full DJFM time period can often be unfairly biased by the high occurrence of zeros, which artificially decreases the average differences and standard deviations. To allay this issue such that we fairly compare NH $A_{NAT}$, we modify our analysis procedure as follows: We use time series of MERRA-2 NH $A_{NAT}$ from November through April on 30, 50, and 70 hPa to define approximate start and

end dates for the periods having non-zero $A_{NAT}$. We use 30 hPa to define the onset dates (because $A_{NAT}$ usually first becomes nonzero around this level; e.g., Figure 7a), and 50 hPa or 70 hPa to define the termination dates. More specifically, we define the onset dates for each year as the first day at 30 hPa having nonzero $A_{NAT}$, and the termination dates as the latest day chosen by either 50 or 70 hPa having nonzero $A_{NAT}$; both 50 and 70 hPa are used because termination most often happens latest around 70 hPa as seen in Figure 7a, but in some winters it happens later around 50 hPa. This process gives us "NAT seasons" between

1979/1980 and 2016/17 having a median length of 87 days, with the minimum and maximum number of days being 41 and 126, respectively. We then use these truncated time series to define the average differences and standard deviations thereof. This modifies the bootstrapping procedure described in Section 2.2.3; we still perform $10^5$ stationary bootstraps for each year, but because the lengths of the time series vary, we also vary the expected block size for each year by specifying them as the nearest integer to the cube root of the time series lengths (which ranges from 3 to 5 for time series lengths between 41 and 126

days).

Figure 8 shows $A_{NAT}$ differences for SH winter seasons. As was the case for the minimum temperatures, there is a large decrease in both the magnitude of the average differences and the standard deviations after 1998. The changing signs of the differences with height before 1999 are also consistent with those seen in the minimum temperatures, with positive differences between about 50 and 20 hPa sandwiched between negative differences above and below for CFSR, ERA-I, and JRA-55.

Furthermore, the increase in average differences in the upper levels between 1993 and 1998 is again apparent. The average differences generally show more regions that are not significantly different from zero in the later years, though the patterns of this close agreement are rather different than those for the minimum temperatures. For this diagnostic, ERA-I continues to show significant average differences for most levels/regions throughout the period, though the magnitudes of the differences are much smaller than those in the earlier years. The differences that are seen between the patterns of $T_{min}$ versus $A_{NAT}$



average differences suggest some minor differences between reanalyses in the morphology of the fields (e.g., spatial patterns or gradients) beyond just overall temperature biases.

$A_{NAT}$ differences in the NH (Figure 9), like the corresponding minimum temperature differences, show more complex patterns than those in the SH and less of an obvious convergence toward better agreement after 1999. The differences do decrease
after about 1999 to 2000, with magnitudes of most differences below about 0.3% of a hemisphere after 2000. JRA-55 shows a band of slightly larger differences in the later years between about 30 and 15 hPa. CFSR and ERA-I (and to a lesser degree, MERRA) show an increase in average differences and standard deviations between about 1994 and 1998. As was the case for the SH, the standard deviations decrease over time. CFSR and ERA-I have larger standard deviations (indicating greater variation in the differences during each year) than JRA-55 and MERRA in the first approximately half of the comparison period, but
show significant decreases after about 2000 to 2001 such that they become comparable to those for the other reanalyses. These patterns of differences are consistent with those seen in $T_{min}$. While the average differences are small throughout the comparison period, they are often significantly different from zero, and there is not in general a clear trend towards less significance. The patterns of significant average differences are distinctly different than those for minimum temperatures.

## 3.2  Vortex Diagnostics

Figure 10 shows the NH and SH climatologies of MERRA-2 MPVG. The evolution of MPVG is quite similar in both hemispheres, particularly above 500 K; the gradients in sPV gradually increase over time, reaching maxima in roughly mid-Feb in the NH and early Oct in the SH. These patterns largely reflect two competing effects: one is the seasonal cycle of the vortex building up strength and subsiding. The other is the build-up effect from wave breaking and mixing/erosion of PV in the surf zone (the region of low-magnitude PV outside the vortex, e.g., McIntyre and Palmer, 1984) over the season, which can act to
sharpen the gradients of PV in the vortex edge region. In the absence of large disturbances, large MPVG indicates the strength of the vortex edge as a barrier to transport. For simplicity, in the discussion of results below, we will refer to $1.0 \times 10^{-6}$ s$^{-1}$ deg$^{-1}$ as 1 scaled PV gradient unit, or 1 PVGU.

The averages and standard deviations of differences from MERRA-2 SH MPVG are shown in Figure 11. All the reanalyses show similar patterns of differences from MERRA-2; particularly in CFSR, ERA-I and JRA-55 before roughly 1999, there
is one band of positive differences on the order of $2 - 4$ PVGUs between roughly 490 and 580 K that is sandwiched between negative differences at levels above and below. This positive band seems to be slightly higher in MERRA, ranging from roughly 550 to 660 K. The standard deviations in these positive bands are relatively small (usually on the order of $1 - 2$ PVGUs), indicating that the reanalyses tend to have systematically larger MPVG at these levels and times. The standard deviations also show that the variances of the differences tend to increase with height, especially at levels in the middle stratosphere
above 700 K where the differences often exceed $2.5 - 3$ PVGUs. There is a noticeable shift toward better agreement after 1999 similar to that seen in the SH temperature diagnostics; many of the average differences are near zero, and relatively few of the average differences at different levels and years after 1999 are significantly different from zero. This is also reflected in the standard deviations, which markedly decrease, especially at levels in the lower stratosphere below 660 K. As for the





temperature diagnostics, the TOVS to ATOVS transition most likely played a large role in this shift, with differences in the handling of this transition and the addition of AIRS radiances in 2002 also expected to be significant factors.

In contrast to the temperature diagnostics shown in Section 3.1, the magnitudes of MERRA − MERRA-2 MPVG differences are generally as large as or larger than those for the other reanalyses. Two likely reasons for this are the improvement in

MERRA-2 polar winter PV fields from using the cubed sphere grid (Gelaro et al., 2017) and the fact that MERRA provided PV only on a reduced resolution grid and on pressure levels with much coarser spacing than the model levels.

For MPVG in the NH (Figure 12), there are no predominant patterns among the reanalyses. MERRA and JRA-55 both tend to have larger MPVG than MERRA-2 by 1 − 3+ PVGUs across most of the levels and years, while CFSR tends to be smaller at most levels and years by 1 − 2 PVGUs. The ERA-I average differences look relatively similar to those for the SH with a

positive band between 520 and 580 K, but overall the differences are much smaller than those in the SH, and many more are not significantly different from zero. The standard deviations of the differences are largely consistent between the reanalyses, with values that increase consistently with height from less than 0.8 PVGUs at the lowest levels, to above 1.5+ PVGUs. All of the reanalyses show signatures of several years between 1994/95 and 2000/01 having larger magnitude average differences and standard deviations at levels above 620 K, as was the case for the temperature diagnostics shown in Section 3.1. There

is some indication of a convergence toward better agreement after roughly 2002, when the reanalyses (except JRA-55) began assimilating AIRS radiances; (Figure 1). This is particularly clear in ERA-I, for which most of the average differences are not significantly different from zero over all of the levels. This pattern is also apparent in CFSR and MERRA, but not for JRA-55; the JRA-55 differences from MERRA-2 do not seem to noticeably change after 2002.

Figure 13 shows the MERRA-2 climatologies of SVA for both hemispheres. Similar to MPVG, the seasonal patterns of SVA

for both hemispheres are quite similar. In this case, the patterns are largely due to the lack of sunlight early in the winter season, which gradually returns later on. However, there are notable differences between the hemispheres, particularly that SVA tends to be smaller in the NH; this is because the NH polar vortex is almost always smaller than its SH counterpart. During individual winters, and given sufficiently low temperatures, the amount of vortex air exposed to sunlight at any time is generally indicative of the amount of air where ozone depletion can take place.

The averages and standard deviations of differences from MERRA-2 SH SVA are shown in Figure 14. Here, the average differences are relatively consistent between the reanalyses, with the four reanalyses generally having bands of positive average differences (indicating the reanalyses having higher SVA) on the order of 1 − 1.5% of a hemisphere in between about 520 to 700 K. These bands of positive differences are consistently largest and widest for MERRA and ERA-I. The standard deviations of the differences are the highest at levels above 660 K where they are usually above 1% of a hemisphere (and often above

1.5 − 2% of a hemisphere); they are also relatively large at the lowest levels around 390 and 410 K, which is around the top of the subvortex region for the SH. A shift towards better agreement is seen most clearly in CFSR, ERA-I, and JRA-55 after 2000; this shift is particularly evident in the standard deviation of differences, which become less than 0.4% of a hemisphere at most of the levels between 390 and 850 K. MERRA is the only reanalysis that does not show as marked of a shift toward better agreement with MERRA-2 − particularly at 660 and 700 K, MERRA SVA is systematically higher than SVA from MERRA-2.





Investigation of the reanalyses' differences in vortex area from MERRA-2 reveal they are nearly identical to the ones for SVA. This indicates that the differences are largely dominated by differences in the size of the vortex edge contours.

Figure 15 shows the same averages and standard deviations of differences in SVA, but for the NH. In this case, each of the reanalyses show different patterns of differences from MERRA-2: CFSR has a band of positive differences greater than 1% of a hemisphere at the lowest levels from roughly 410 to 490 K; ERA-I generally has small differences less than 1% of a hemisphere at all levels except at levels around 800 to 850 K; JRA-55 also has generally small negative differences above -1% of a hemisphere; and MERRA has a band of positive differences greater than 1 – 1.5% of a hemisphere between roughly 660 and 800 K. These patterns of average differences do not noticeably change much over the full range of years, suggesting they are fairly insensitive to jumps in the observing system. The standard deviations of differences from MERRA-2 SVA all look consistent between the reanalyses, with the largest values greater than 1 – 1.5% of a hemisphere usually confined to a band of levels between 660 to 850 K. At lower levels, however, the standard deviations are quite small throughout the period, generally on the order of 0.5% of a hemisphere or less. Similar to SH SVA, an investigation of the reanalysis differences in vortex area from MERRA-2 also reveals that the differences are nearly identical to the ones shown in Figure 15. Thus, while there is no consistent change in agreement over the years, these results indicate persistent differences in the size of the contours used to define the vortex edges, and hence some persistent differences in the isentropic PV fields.

### 3.3 Derived Vortex-Temperature Diagnostics

The diagnostics shown in the following subsection are derived from the temperature and/or vortex diagnostics shown in the previous two subsections.

The number of days with temperatures below $T_{PSC}$ summed over lower stratospheric levels has been previously used as a summary measure of the extent and duration of the period conducive to polar processing (e.g., Manney et al., 2011, 2015; Lawrence et al., 2015). Figure 16 shows the total days with $T < T_{ice}$ in the SH during each winter, calculated for the "central" PSC threshold and the $\pm 1$ K sensitivity thresholds (see Section 2.2.2), over pressure levels from 121.1 to 31.6 hPa. The total number of SH days with $T < T_{ice}$, calculated from the central values, ranges from about 700 (in 1981 in MERRA and CFSR, and in 2002 in JRA-55) to over 1050 (in 1999, 2006, and 2015, and in some reanalyses in 1987 and 1996), thus showing significant interannual variability in temperatures summed over the lower stratosphere. (Note that all of the years with largest values except 1996 are years with very deep ozone holes, e.g., WMO, 2014; Nash et al., 2016). Figures 16b and d show that these values could vary by between nearly 100 and nearly 200 days depending on the exact temperature used for the PSC threshold. The central values vary among the reanalyses by anywhere from nearly 80 days (e.g., 1980) to less than 20 days (e.g., 2006, 2011). Most of the years with the largest differences among the reanalyses are in the 1980s, with 2002, 2004, and 2012 being the only years with a spread among the reanalyses greater than 30 days after 1997. Thus there does seem to be some convergence – though not monotonic – towards better agreement among the reanalyses. There does not appear to be a consistent order of the reanalyses – the only significant pattern seems to be that ERA-I is near the low side in the period since 1998. The MERRA and MERRA-2 bars often are not next to each other, suggesting that agreement between the two is not





consistently better than that among the other reanalyses. This suggests that the details of the agreement among the reanalyses depend strongly on the details of the meteorological conditions in a given year.

Figure 17 show the total number of days in the NH lower stratosphere with $T < T_{NAT}$ each winter. The interannual variability is, of course, much larger here than in the SH, and the number of days with $T < T_{NAT}$ often much smaller than that with $T <$

$T_{ice}$ in the SH. Central values range from about 120 to 250 in many of the years with strong/prolonged SSWs in December or January (1984/1985, 1998/1999, 2001/2002, 2003/2004, 2008/2009, 2012/2013) (e.g., Manney et al., 1999, 2005b, 2009, 2015; Naujokat et al., 2002). The range of values that might be seen based on the uncertainty in the PSC threshold temperature (Figure 17b and d) varies from slightly more than 100 days (e.g., 1987/1988, 2005/2006, 2012/2013) to over 300 days (e.g., 1993/1994, 2014/2015), and does not correlate strongly with the total number of days. This can be a up to about 50% of the

total number of days in some winters. The central number of days in an individual years varies by anywhere from 11 days (e.g., 2008) to around 80 days (e.g., 1993/1994, 1994/1995, 1995/1996), with a cluster of years from 1992/1993 to 1997/1998 with large spreads among the reanalyses; the other large spreads between reanalyses are in 2013/2014 and 2014/2015, with most other years showing spreads between about 20 and 40 days; thus, there does not appear to be any pronounced convergence toward better agreement over the years. After about 1989, CFSR is usually near the top of the range (appearing at or near the

right side of the year columns); the other reanalyses don't show obvious preferred positions. In 2011 and 2016 (the latter for the three reanalyses, ERA-I, JRA-55, and MERRA-2, that are available as we write this), the overall coldest Arctic winters in the record since 1979, the reanalyses agree very well, but not all cold winters show such close agreement (e.g., 1995 and 1996). While there are some fairly large differences and sensitivities among the reanalyses, all of the reanalyses do show similar interannual variations among the NH winters.

The winter mean volume of lower stratospheric air with temperatures below $T_{PSC}$ ($V_{PSC}$) is a widely used diagnostic of polar processing potential, and it is often expressed as a fraction of the vortex volume (e.g., Rex et al., 2004, 2006; Tilmes et al., 2006; Manney et al., 2011; Manney and Lawrence, 2016; WMO, 2014). Here, we calculate $V_{PSC}$ and the volume of the vortex using $A_{PSC}$ and $A_{Vort}$ on isentropic levels between 390 and 550 K by assuming each isentropic level is nominally representative of the volume of air midway between each level; for example, the 410 K level comes after 390 K and before 430 K, so 410 K is

assumed to be representative for altitudes between 400 and 420 K. The altitudes for these nominal levels are determined using the Knox (1998) approximation; for the levels from 390 to 550 K, this gives a mean altitude differential of 1.13 km with a range of 0.98 – 1.30 km. These altitude differentials are then multiplied by the area diagnostics on each isentropic level (which are converted to km$^2$), and summed over the vertical range to get volumes. The volume fraction is then $V_{PSC}/V_{Vort}$.

Figure 18 shows the winter mean volume of temperatures below $T_{ice}$ in the SH expressed as a fraction of the vortex volume,

in the same format as in Figure 16. Keeping in mind that $T_{ice}$ was estimated assuming nominal pressure levels for isentropic levels (see Section 2.2.2), which results in a significant overestimate of areas/volumes, Figure 18 shows that the volume fraction of cold air is relatively constant from year to year. Generally, the fractions of the vortex are between 0.15 and 0.25 each year, with sensitivities to the ice threshold offsets between roughly 0.05 and 0.075. Between the winters from 1979 and 1986, there is a very persistent pattern with CFSR having the lowest, and ERA-I having the highest, cold volume fractions of the vortex.

During this period, CFSR can be lower than the other reanalyses by nearly 0.025 to 0.03. After these years, for nearly all



years between 1995 and 2016, ERA-I shows the lowest volume fractions ranging from roughly 0.01 – 0.02 lower than the other reanalyses. For years from 2009 to 2015, JRA-55 consistently has the highest volume fractions, but in these cases the differences from the other reanalyses are generally quite small. Differences in the temperature threshold sensitivity envelopes among the reanalyses do suggest some minor differences in horizontal temperature gradients, but nothing overtly persistent.

Potential polar processing volumes in the NH are much lower and much more variable than those in the SH. The NH fraction of vortex volume below $T_{NAT}$ (Figure 19) shows values in the colder years that are comparable to those below $T_{ice}$ in the SH. The lowest values are seen in 1984/1985, 1998/1999, 2001/2002, and 2003/2004, all years with very early (mid-December to the beginning of January) major SSWs that profoundly affected the entire stratosphere, including strongly disrupting the lower stratospheric vortex (e.g., Manney et al., 1999, 2005b; Naujokat et al., 2002); in these years the fractional volume is
near 0.03, as opposed to nearly 0.30 in the coldest years (e.g., 1996, 2011, 2016). The range of values from the PSC threshold temperature sensitivity tests varies from about 0.03 in the warmest years up to nearly 0.10 in the coldest years, with differences betweeen reanalyses indicating some differences in horizontal temperature gradients (especially in, e.g., 2011 and 2014). The interannual variability is well represented in all of the reanalyses. The central values usually vary more between reanalyses in colder years – e.g., 2011 stands out as showing a very wide range of about 0.06. As in the SH, through 1989 CFSR stands out
as having the lowest NAT volumes. CFSR and ERA-I are among the lowest during most of the record. JRA-55 typically has the largest NAT volumes during most of the period examined, with only MERRA-2 being slightly larger in 2001/2002 through 2003/2004 and 2005/2006. While many of the recent years show smaller ranges of central values than the early years, there is not a monotonic progression, so any trend towards better agreement is masked by the larger influence of specific interannually varying conditions that affect the PSC volumes.

The SH vortex breakup is of considerable concern because it results in the dispersal of ozone-depleted vortex air over mid-latitudes (e.g., Ajtić et al., 2003, 2004; Manney et al., 2005c; Pazmino et al., 2005; WMO, 2007). While ozone depletion in the Arctic has not yet been large enough for this to be an ongoing concern, vortex evolution during the 2011 Arctic vortex breakup led to significant areas of ozone depleted air over populated regions associated with increased surface UV (e.g., Manney et al., 2011; Bernhard et al., 2012). To examine the variability and representation in reanalyses of the vortex breakup in the lower to
middle stratosphere, we examine approximate vortex decay dates, which we derive from the vortex area diagnostic using the $+0.1 \times 10^{-4}$ s$^{-1}$ offsets on isentropic levels from 460 to 850 K. To accomplish this, we examine NH $A_{Vort}$ between 1 Dec and 1 Jun, and SH $A_{Vort}$ between 1 May and 1 Mar; we have defined the decay date as the last day before which $A_{Vort}$ is above 2% of a hemisphere continuously for 30 days. We choose 2% of a hemisphere as the limit because it is well below the NH DJFM and SH MJJASO $A_{Vort}$ climatologies at all levels, which guarantees that any time the vortex is that small, it is either significantly
disturbed or in the process of decaying. The 30 day limit is chosen to help guarantee that the vortex was sufficiently coherent beforehand. Our results are not highly sensitive to changing the area threshold or using vortex area with/without the sPV offset; except in some marginal cases (discussed below), adjusting the area threshold between 2 and 4% only modifies the decay dates by less than 12 days in the NH, and less than 5 days in the SH.

Figures 20 and 21 show pixel plots of the MERRA-2 vortex decay dates and the differences from the other reanalyses (i.e.,
the reanalyses minus MERRA-2). For the SH, Figure 20 shows that the vortex tends to decay fairly late in the year, and it



does so earlier at the upper levels than at the lower levels; in other words, the vortex in the SH typically decays from the top down. The differences in decay dates among the reanalyses are generally less than 2 weeks, and in most cases are between -4 and 4 days. All the reanalyses show similar patterns, at least between 1979 and 1999, with a band of positive differences for levels from roughly 580 to 660 K sandwiched between bands of negative differences at the top and bottom levels. After 1999, these positive differences seem to expand upward to higher levels; this is especially the case for MERRA, which shows some of the largest positive differences in decay dates from MERRA-2 after 2000. These results seem generally consistent with the differences in SVA shown in Figure 14, with positive (negative) differences in decay dates in the same regions there are positive (negative) differences in vortex area. There are a few cases with very large positive differences, particularly in 2002 and 2009, that show up in MERRA, JRA-55, and CFSR. In the 2002 SH winter, a major SSW and vortex split led to the vortex breaking down at levels above 850 K by mid-October; in MERRA-2, the vortex area oscillated above and below 2% of a hemisphere at 850 K, whereas in JRA-55 and MERRA it stayed above 2% consistently for more than 2 extra weeks. Although there was no SH SSW in 2009, similar marginal conditions occurred at 850 K late in the season, leading to large differences in the decay dates in CFSR, JRA-55, and MERRA.

Figure 21 shows that the NH vortex breakup is much more variable from year to year than that in the SH. Unlike the SH vortex, the NH vortex can decay nearly simultaneously over a wide range of levels (e.g., 1984 and 1999), or it can decay earlier at some low levels, and later at higher levels (e.g., 2001 and 2009). Such variability in vortex decay is due to large variability induced by SSW disturbances to the vortex, as well as polar night jet oscillation events in which the middle and upper stratospheric vortex rapidly reforms following some major and minor disturbances (e.g., Hitchcock et al., 2013; Lawrence and Manney, 2017, and references therein). The reanalyses' differences from MERRA-2 are generally quite small, usually within -2 to 2 days. There are no predominant patterns of differences (e.g., positive or negative bands), but there are many more outliers with absolute differences from MERRA-2 greater than 20 days (denoted by the white x symbols). Many of these cases are marginal scenarios when the MERRA-2 vortex area hovers above and below the specified 2% of a hemisphere threshold at some levels, causing our algorithm to pick a much earlier decay date than the other reanalyses that (in comparison to MERRA-2) persistently stay above 2% at the end of the season. Similar to the SH decay dates, some of the persistent differences among the reanalyses in NH decay dates are due to the persistent differences in vortex area discussed in Section 3.2; CFSR tends to have higher vortex area than MERRA-2 between roughly 430 and 490 K, at the same levels CFSR has some of the largest (non-outlier) decay date differences. MERRA also shows this behavior at levels between 660 and 800 K.

## 4 Conclusions

We have herein done an extensive intercomparison of diagnostics of polar chemical processing among the five most recent full-input reanalyses, using the most recent, MERRA-2, as a reference to compare MERRA, ERA-I, JRA-55, and CFSR. The diagnostics we compare are based on polar vortex and temperature conditions in the lower stratosphere, and comprise measures of PSC formation and chlorine activation based on temperatures; vortex size, strength, and sunlight exposure; and additional diagnostics derived from those directly obtained from temperatures and vortex characteristics. They thus provide a



thorough assessment of the reanalyses' representation of the potential for polar processing and ozone loss in both hemispheres. Compared to previous studies, we include all of the latest generation reanalyses, examine the sensitivity of the diagnostics to uncertainties in temperature and vortex threshold values used, and provide an assessment of the statistical significance of the differences between reanalyses. The main findings of our analyses are summarized in the following subsection.

## 5  4.1  Summary

Temperature diagnostics converge towards better agreement in the SH over the period compared (from 1979 to present), with agreement in minimum temperatures generally within about 1 K after 1998 (as opposed to up to about 6 K before) and largest $A_{NAT}$ differences decreasing from near 2% to less than about 0.5% of the hemisphere. A large sudden decrease in both the reanalysis differences and the standard deviations thereof is seen in 1999, consistent with previous studies (e.g., Long et al.,
2017) that show large improvements in zonal mean temperatures after the TOVS/ATOVS transition. In the NH, the agreement before 1998 was already much closer (within about 2 K for minimum temperatures), but differences and standard deviations do decrease to some extent over the years. Differences in both hemispheres show a banded structure with height, with MERRA-2 generally being warmer than the other reanalyses between about 50 and 30 hPa; the standard deviations of the differences generally increase with height. Between about 1994 and 1999, increased differences and standard deviations are seen above
about 30 hPa in both hemispheres, which may be related to increasing impacts of differences in how the data are assimilated. In both hemispheres, temperatures diagnostics before 1998 show closer agreement between MERRA-2 and MERRA than between MERRA-2 and the other reanalyses.

Differences among the reanalyses for SH maximum PV gradients are generally similar, with banded structures of positive and negative differences from MERRA-2 (particularly in CFSR, JRA-55, and ERA-I), and standard deviations that increase with height. The differences from MERRA-2 in these cases are generally within $\pm 4$ PVGUs prior to 1999, but only roughly $\pm 1$ PVGU thereafter. For NH maximum PV gradients, there are no consistent patterns of differences from MERRA-2 among the reanalyses, but, similar to the SH, the standard deviations of the differences tend to increase with height. Generally the differences are within $\pm 2$ PVGUs over all the years; however, there is a noticeable convergence in agreement between ERA-I and MERRA-2 after 2002 that is not as apparent in the other reanalyses. Differences from MERRA-2 in sunlit vortex area gen-
erally follow differences in vortex area itself. In the SH these average SVA differences can be as large as 2% of a hemisphere, but they decrease in magnitude (in both the averages and standard deviations) after roughly 1999. In the NH the average differences are generally small, except for some persistent positive average differences in limited bands that show up in MERRA between roughly 660 and 750 K, and in CFSR between roughly 410 and 490 K. For these PV-based diagnostics, MERRA and MERRA-2 show differences of magnitude as large as (sometimes larger than) those between MERRA-2 and the other reanaly-
ses; the reduced resolution PV provided for MERRA and the cubed-sphere grid used for MERRA-2 are likely factors in these differences.

The derived winter summary diagnostics, which include the number of days (summed over vertical levels) below PSC thresholds, the volume of air below PSC thresholds (expressed as a fraction of the vortex volume), and vortex decay dates, generally agree better in the SH than in the NH. Particularly for the number of days below $T_{ice}$ and $V_{ice}$ for the SH, all the




reanalyses show similar magnitudes and sensitivities to the $\pm 1\,\mathrm{K}$ temperature offsets. For the NH, the number of days below $T_{NAT}$ and $V_{NAT}$ vary much more from year to year, and the differences among the reanalyses and sensitivities to the temperature offsets are much larger percentages of the actual derived values. These characteristics are in many ways to be expected, since SH winters are much more consistent from year to year than NH winters; thus, even though the individual polar processing

diagnostics show much larger average differences and standard deviations for the SH, the aggregation of the full winter seasons done for the derived diagnostics leads to more consistent results. These findings are also consistent with the vortex decay dates, which, except in rare cases, generally vary by less than a week among the reanalyses for the SH. While the differences in decay dates are also often quite small in the NH, the early vortex breakup (relative to the SH) and the frequent occurrence of midwinter SSWs and significant vortex disturbances make large differences more common because of more frequent marginal

cases.

## 4.2 Implications

The results shown herein illustrate some implications that may be expected for polar processing studies using reanalysis temperatures and PV in the stratosphere. These implications will generally depend on the hemisphere in question and the detail of the study. For example, the derived diagnostics in Section 3.3 demonstrate that in the aggregate most SH winters

in the satellite era are quite similar, and that the sensitivities to different PSC temperature thresholds are consistent among the reanalyses. However, the differences shown in Section 3.1 indicate that differences can depend strongly on the levels and years examined, especially prior to 1999 before the assimilation of AMSU data in the reanalyses. Thus, studies that discuss SH winter conditions in aggregate are less likely to be affected than detailed studies (e.g., those making use of nudged and specified dynamics models, and/or Lagrangian transport models), whose conclusions could be significantly altered by the details of how,

when, and where the temperatures differ among the reanalyses. In contrast, for the NH, Section 3.1 showed that temperature diagnostic differences were relatively small among the reanalyses, but the results in Section 3.3 showed that the aggregate derived diagnostics vary widely between reanalyses in some cases, and can be highly sensitive to the specific temperature thresholds used. Clearly polar processing potential is often much smaller in the NH than in the SH, and thus conclusions based on the often marginal conditions of the NH are much more likely to be affected by small differences among the reanalyses.

Thus, both detailed and aggregate studies of NH polar processing could in some cases be markedly affected by differences among the reanalyses. However, all of the reanalyses do show similar interannual variations among the derived diagnostics, and thus for purposes of putting some NH winters into the context of others (e.g., comparing how cold some are relative to others), any of the reanalyses would give similar results. The extent to which different kinds of studies of NH and/or SH polar processing may be affected is beyond the scope of this paper, but work is in progress within S-RIP to explore some of these

implications.

It is difficult to assess the potential implications of differences among the reanalyses in the vortex diagnostics. Since MERRA-2 is the only reanalysis that provides PV fields on its model levels, we have applied the strategy we think other data users requiring PV on model levels would use, which was to derive PV from each reanalysis using their available model level products. Thus, it is important to recognize that the vortex diagnostics used herein are derived from PV fields that are





calculated from the different reanalyses in different ways, which makes it problematic to assess whether and the extent to which the reanalysis differences are due to differences in calculations, dynamics, vertical and/or horizontal resolution, etc. Because MERRA-2 includes PV calculated within its DAS, we generally consider MERRA-2 PV to be more consistent and complete than the PV fields derived from the other reanalyses' model level data. Significant differences between MERRA-2 and MERRA

PV-based diagnostics suggest that resolution may be a significant factor (since the MERRA PV was also calculated within the DAS, but was only available on a reduced resolution grid), but model changes may also play a role. Despite these complicating factors, our treating each of the reanalyses equally (same procedure for calculating MPVG, and same contours used to define vortex area) allows us to draw some useful conclusions: While there were some indications of convergence toward better agreement in maximum PV gradients and sunlit vortex area for both hemispheres among some reanalyses (see Section 3.2),

there were persistent differences elsewhere. Given the combination of differences in MPVG and SVA, these results indicate that using the same vortex edge values for each reanalysis is not always be an appropriate simplification. Differences among the reanalyses in MPVG alone indicate that there may be differences in the equivalent latitude mapping of the PV fields (which, again, could arise for numerous reasons), which in some cases could alter conclusions drawn about transport barriers and trace gases in equivalent latitude coordinates.

## 4.3 Recommendations

All of the reanalyses used here represent vast improvements over those commonly used a decade ago, and with those improvements comes much closer agreement in the polar processing diagnostics presented here. The older reanalyses, especially ERA-40 and NCEP/NCAR and NCEP/DOE, have long been obsolete and are not recommended for studies focused on polar processing and the stratosphere in general (see Fujiwara et al., 2017, and references therein). Any of the modern reanalyses

evaluated herein are much better choices for polar processing studies as they all provide more accurate and similar representations of interannual variability in polar processing diagnostics in both hemispheres.

In general, it is always better to use more than one reanalysis, even for studies involving recent winters where it can reasonably be expected that differences among the reanalyses will be small. One of the best ways to express uncertainty in results is using multiple reanalyses, and explicitly showing and discussing how they agree/disagree, and whether any differences affect

the findings; this is especially important for diagnostics that cannot be compared with observations. As previously shown by Lawrence et al. (2015) for polar processing diagnostics, and Long et al. (2017) for zonal means, our intercomparisons (see particularly Figures 5 – 6 and 8 – 9) show that there are substantial (especially large in the SH) changes in temperature-based diagnostics that are clearly related to changes in assimilated data inputs among the reanalyses. Since many of the major changes in data inputs are made at approximately the same time in each reanalysis, the agreement or lack thereof between the reanal-

yses does not provide the information to assess the degree to which these changes are caused by changes to the assimilated observations. We thus emphasize here that reanalysis temperatures, especially in the Antarctic, are not generally suitable for assessment of trends in temperature-based diagnostics; use of reanalyses in trend studies should be regarded with skepticism and only attempted with the use of multiple reanalyses, and after rigorous assessment of the relationships of temperature changes to observations assimilated (which, to our knowledge has not been done).



When using multiple reanalyses, it is important to treat them as fairly and equally as possible to reduce the uncertainty in sources of differences. For example, using one reanalyses with data on model levels, and another reanalysis with data on pressure levels is not recommended. It is also important to be clear whether and how fields/quantities are derived from the products provided by the reanalyses, as we have done herein with PV. Until and unless reanalysis centers provide standard sets

of products on standardized isobaric and isentropic levels, users of reanalysis data will generally be best served by using model data to vertically interpolate and derive fields as needed. Numerous evaluations of reanalyses for S-RIP are finding, as we have here, that it would be valuable to have PV on model levels available in future reanalyses.

With regard to more specific polar processing applications: we also recommend that trends, correlations, and/or other similar analyses of diagnostics that assess low temperatures aggregated over winter months, seasons, and/or vertical levels in the NH

polar region be performed with caution. Figures 17 and 19 demonstrate there is non-negligible interannual variability in the sensitivity to the specific temperature values chosen to represent NAT PSC thresholds that are used to calculate the number of days and volume of air below NAT thresholds in the NH, especially relative to the SH (Figures 16 and 18) in which we used the lower ice PSC thresholds. The vortex diagnostics in Section 3.2 show some differences that appear to be related to biases between PV in the reanalyses, arguing for careful assessment of the sensitivity of vortex diagnostics to exact PV

values. Because many of the diagnostics that are most informative about lower stratospheric polar chemical processing cannot be readily validated by comparison with data, the comparison of reanalyses is a powerful tool for assessing robustness and uncertainty in these diagnostics.

*Data availability.*    The datasets used/produced are publicly available, as follows:

–  MERRA-2: https://disc.sci.gsfc.nasa.gov/uui/datasets?keywords=%22MERRA-2%22

–  MERRA: https://disc.sci.gsfc.nasa.gov/uui/datasets?keywords=%22MERRA%22

–  ERA-I: http://apps.ecmwf.int/datasets/

–  JRA-55: Through NCAR RDA at http://dx.doi.org/10.5065/D6HH6H41

–  CFSR model level data: Available upon request from Karen H Rosenlof (karen.h.rosenlof@noaa.gov)

–  Polar processing diagnostic products: Contact Zachary D Lawrence (zachary.lawrence@student.nmt.edu)

*Author contributions.*    ZDL and GLM designed the study. ZDL did the analysis. KW provided expertise and guidance on the reanalysis datasets. ZDL and GLM wrote the paper. KW read and commented on the paper.

*Competing interests.*    The authors declare no competing interests.





*Acknowledgements.* We thank the Microwave Limb Sounder team at JPL, especially Brian W Knosp and Ryan A Fuller, for computational, data processing, management, and analysis support; Will McCarty and Larry Coy for helpful comments; NASA's GMAO, ECMWF, JMA, and NCEP for providing their assimilated data products; and Amy Butler, Jeremiah Sjoberg, Craig Long, Sean Davis, Henry L Miller, and Karen Rosenlof for processing and providing the model level CFSR data. GLM and ZDL were supported by the JPL Microwave Limb

5   Sounder team under JPL subcontracts to NWRA and NMT; KW was supported by NASA's Modeling, Analysis and Prediction (MAP) program, which also provides support for MERRA and MERRA-2.



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





**Table 1.** List of acronyms of reanalysis assimilated observations

| Acronym | Full Name |
| --- | --- |
| AIRS | Atmospheric Infrared Sounder |
| AMSR | Advanced Microwave Scanning Radiometer |
| AMSU | Advanced Microwave Sounding Unit |
| ATMS | Advanced Technology Microwave Sounder |
| ATOVS | Advanced Tiros Operational Vertical Sounder |
| COSMIC | Constellation Observing System for Meteorology, Ionosphere, and Climate |
| CrIS | Cross-track Infrared Sounder |
| CRTM | Community Radiative Transfer Model (radiative transfer model) |
| GLATOVS | Goddard Laboratory for Atmospheres TOVS (radiative transfer model) |
| GMS | Geostationary Meteorological Satellite |
| GOES | Geostationary Operational Environmental Satellite |
| GPS-RO | Global Positioning System Radio Occultation |
| HIRS | High resolution Infrared Radiation Sounder |
| IASI | Infrared Atmospheric Sounding Interferometer |
| MHS | Microwave Humidity Sounder |
| MLS | (Aura) Microwave Limb Sounder |
| MSU | Microwave Sounding Unit |
| MTSAT | Multi-functional Transport Satellite |
| RTTOV | Radiative Transfer for TOVS (radiative transfer model) |
| SSM/I | Special Sensor Microwave Imager |
| SSMIS | Special Sensor Microwave Imager Sounder |
| SSU | Stratospheric Sounding Unit |
| TMI | Tropical Rainfall Measuring Mission Microwave Imager |
| TOVS | Tiros Operational Vertical Sounder |
| VTPR | Vertical Temperature Profile Radiometer |





**Figure 1.** Time line for operational satellite instrument inputs to the reanalyses used herein: panels (a) through (e) show CFSR, ERA-I, JRA-55, MERRA, and MERRA-2, respectively. Table 1 gives a list of the acronyms used here. Within the constraint of putting them in the same order for each reanalysis, the input datasets are stacked in approximately chronological order, with earliest on the bottom and latest on the top. The black vertical line is at mid-1998, near the time of the TOVS to ATOVS transition (see text).





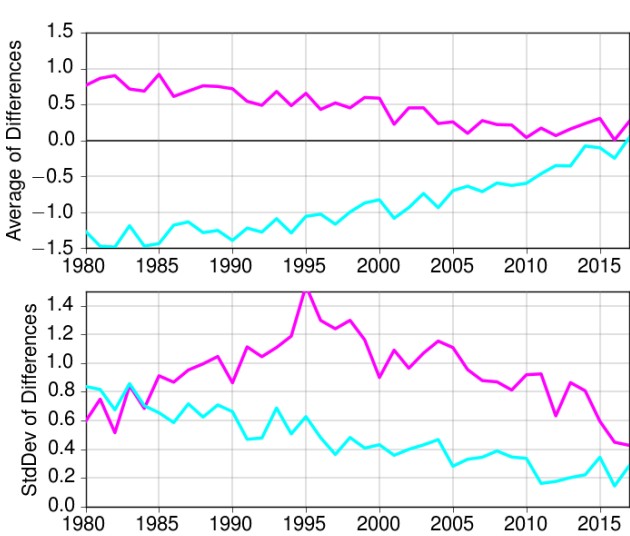

**Figure 2.** Schematic illustration of how "agreement" among reanalyses is assessed. The cyan and magenta lines show (top) the average of the differences between two hypothetical reanalyses and another reanalysis used as a reference and (bottom) the standard deviation of the differences that were averaged. See text for description of how these indicate changes in agreement.





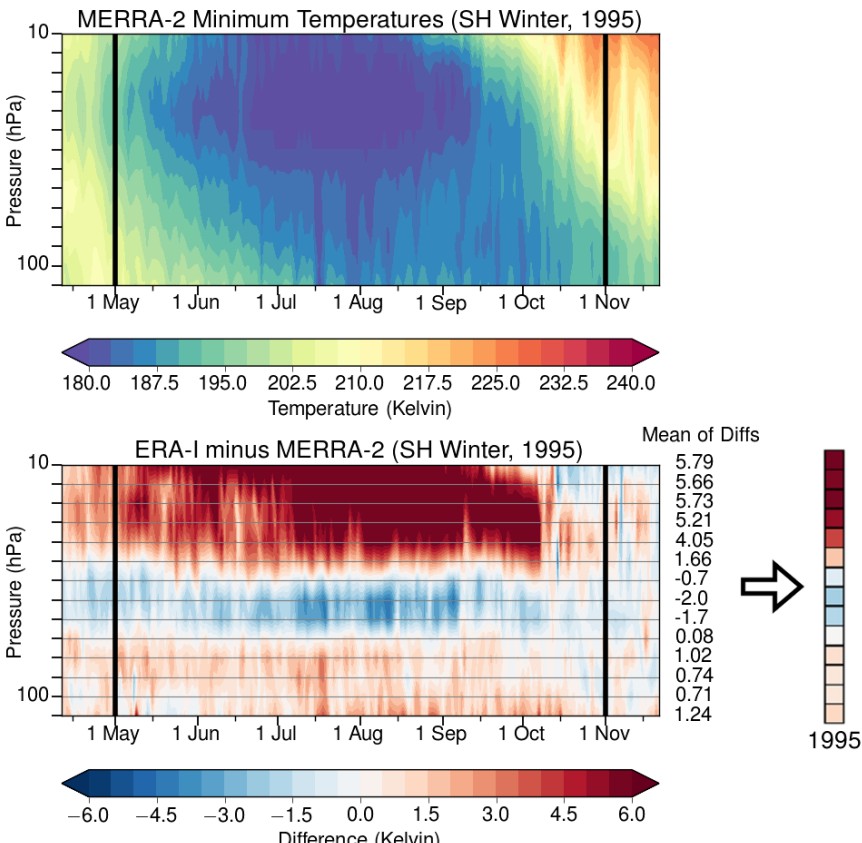

**Figure 3.** Schematic example of steps to go from daily differences to yearly time series. The top panel shows one year (1995) of daily minimum temperatures from MERRA-2; purples/blues (reds) represent low (high) values on the color bar. These values are subtracted from the corresponding ones for each of the reanalyses, yielding fields such as those shown (for ERA-Interim − MERRA-2) in the lower left; here positive (ERA-Interim greater than MERRA-2) differences are shown in reds, negative (ERA-Interim less than MERRA-2) differences in blues. These values are averaged over the period indicated by the black vertical lines to get a number for each level (numbers to the right of difference plot), and those are plotted as a stacked array of squares for each year (lower right).



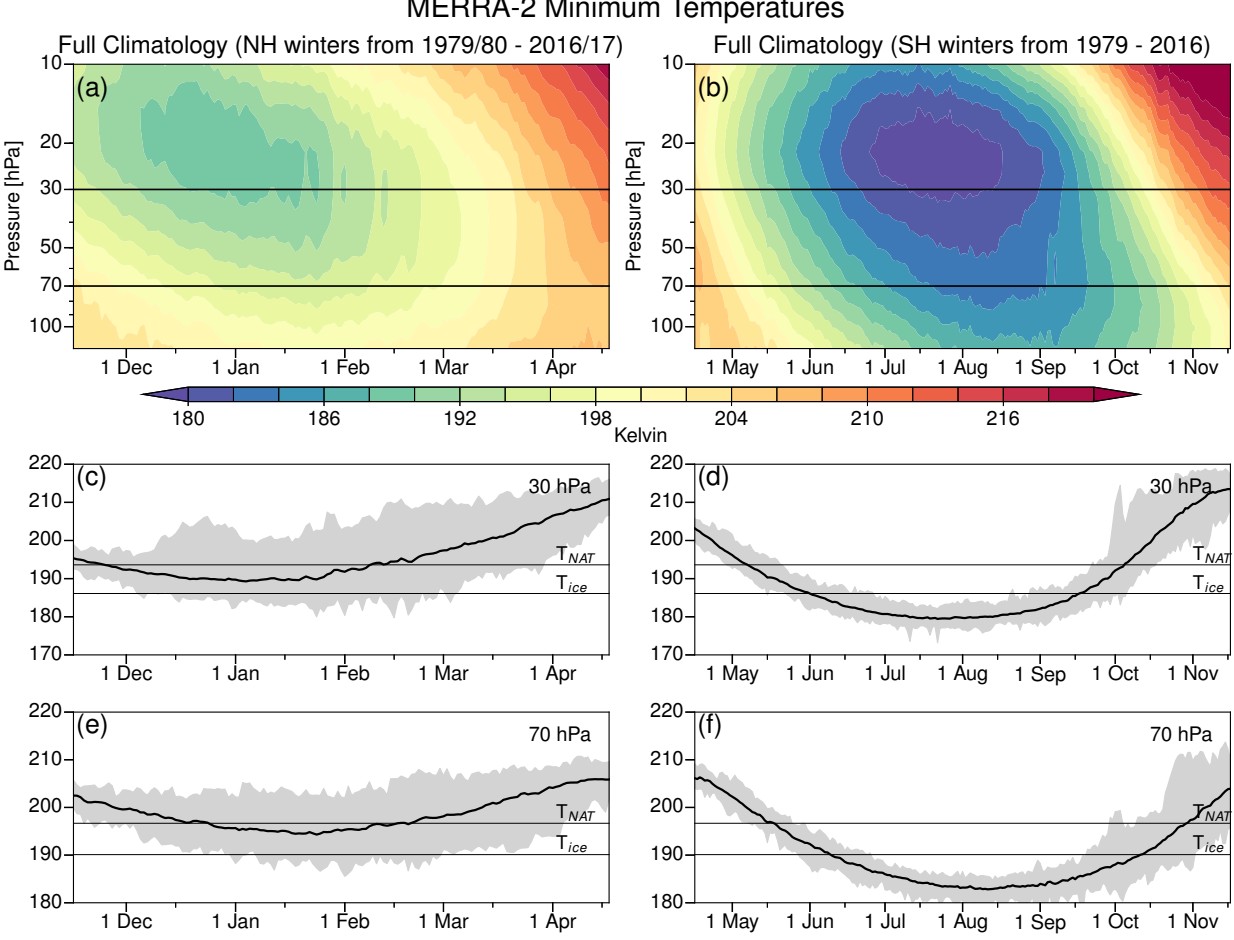

**Figure 4.** Time series of (a, c, e) Arctic and (b, d, f) Antarctic climatological (1979/1980 through 2016/2017 in the NH, and 1979 through 2016 in the SH) minimum high latitude (poleward of 40° latitude) temperatures in the MERRA-2 reanalysis; (a) and (b) show contour plots (blues/purples represent low temperatures and reds high temperatures), with line plots shown at 30 hPa in (c) and (d) and at 70 hPa in (e) and (f). The shading in the lines plots shows the range of values on each date, and the black line the average values. Note that the time period shown is longer in the SH than in the NH. The same color range is used for each hemisphere.





**Figure 5.** SH winter season (MJJASO) (a, c, e, g) averages and (b, d, f, h) standard deviations of minimum temperature differences for each reanalysis from MERRA-2 as a function of year and pressure for the 1979 through 2015 winters, concatenated from the individual years into pixel plots as described in the text and Figure 3. Columns of grey pixels indicate years with no data. Pixels with x symbols inside indicate years and levels where the differences from MERRA-2 are insignificant according to our bootstrapping analysis (see section 2.2.3). Blues in the average difference panels show negative values (reanalysis less than MERRA-2) and reds positive values (reanalysis greater than MERRA-2); in the standard deviation panels, yellows/deep blues represent low/high standard deviations of the reanalysis differences, respectively.





**Figure 6.** As in Figure 5 but for the NH winter seasons (DJFM) for 1979/1980 through 2015/2016. Note that different color ranges are used for the NH shown here than in Figure 5 for the SH.





**Figure 7.** As in Figure 4, but for area with temperatures below the NAT PSC threshold.





**Figure 8.** As in Figure 5 but for area with temperatures below the NAT PSC threshold in the SH.



**Figure 9.** As in Figure 8, but for the NH. See text for explanation of date ranges used for the calculations.



**Figure 10.** As in Figure 4, but for maximum sPV gradients.







**Figure 11.** As in Figure 5, but for maximum sPV gradients.





**Figure 12.** As in Figure 11, but for the NH.





**Figure 13.** As in Figure 10, but for sunlit vortex area.





**Figure 14.** As in Figure 5, but for SH sunlit vortex area.





**Figure 15.** As in Figure 14, but for the NH.



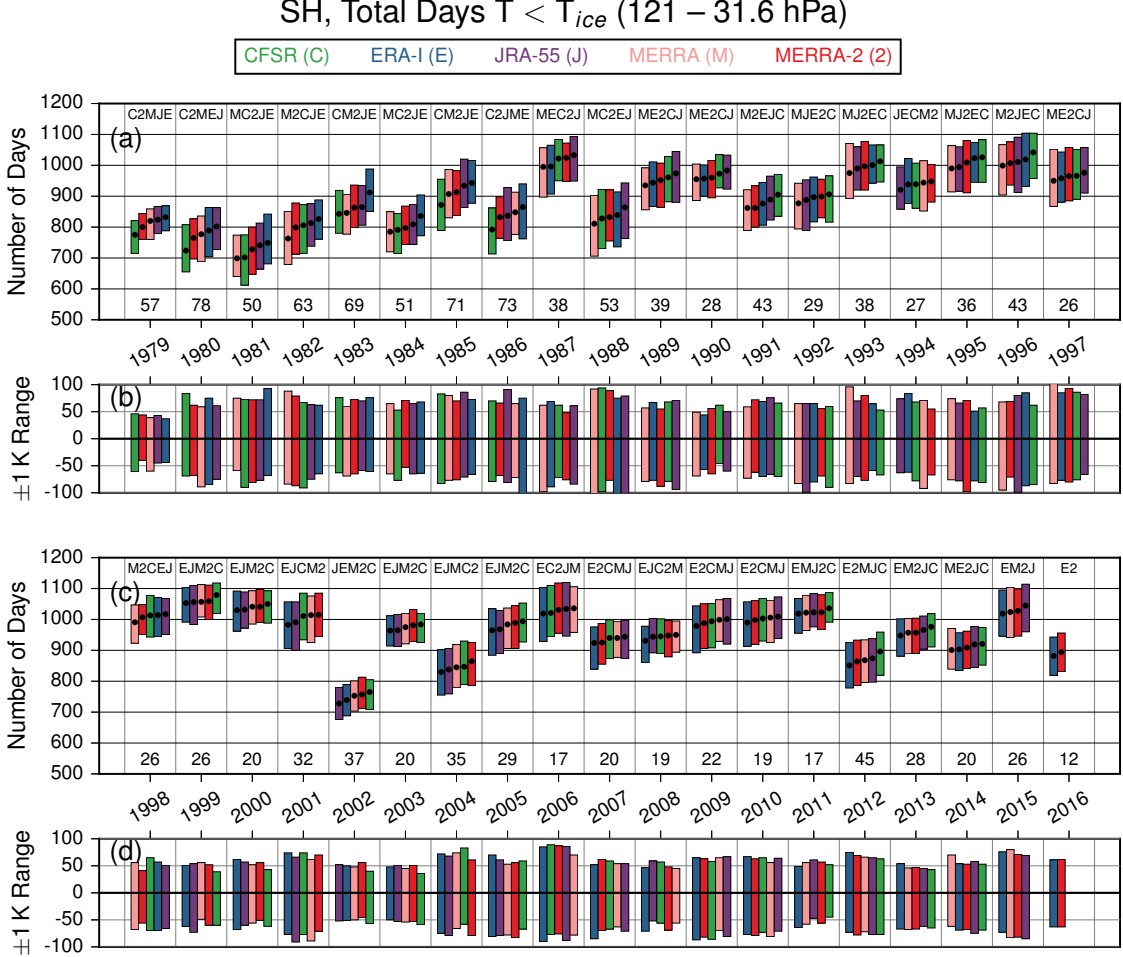

**Figure 16.** Total number of days with temperatures below T$_{ice}$ in the SH for each year summed over the lower stratosphere (from 121.1 to 31.6 hPa) (a and c), and the sensitivity ranges of each reanalysis calculated using ±1 K offsets from T$_{ice}$ (b and d). The colored bars show the range of values obtained from the tests of sensitivity to the PSC threshold temperature used (see Section 2.2.2), while the black dots show the value for the "central" threshold temperature. The days are counted from April through the following February. For each year, the reanalyses are ordered from smallest central value on the left to largest central value on the right; this order is also given as a text string at the top of the column for each year. The numbers at the bottom of each year's column indicate the difference in days between the largest and smallest central values for the year (i.e., between the rightmost and leftmost black dots). In the range panels (b and d), the range about the central value (black dots in a and c) is shown for each reanalysis. Green, blue, purple, pink, and red indicate CFSR, ERA-I, JRA-55, MERRA, and MERRA-2, respectively.





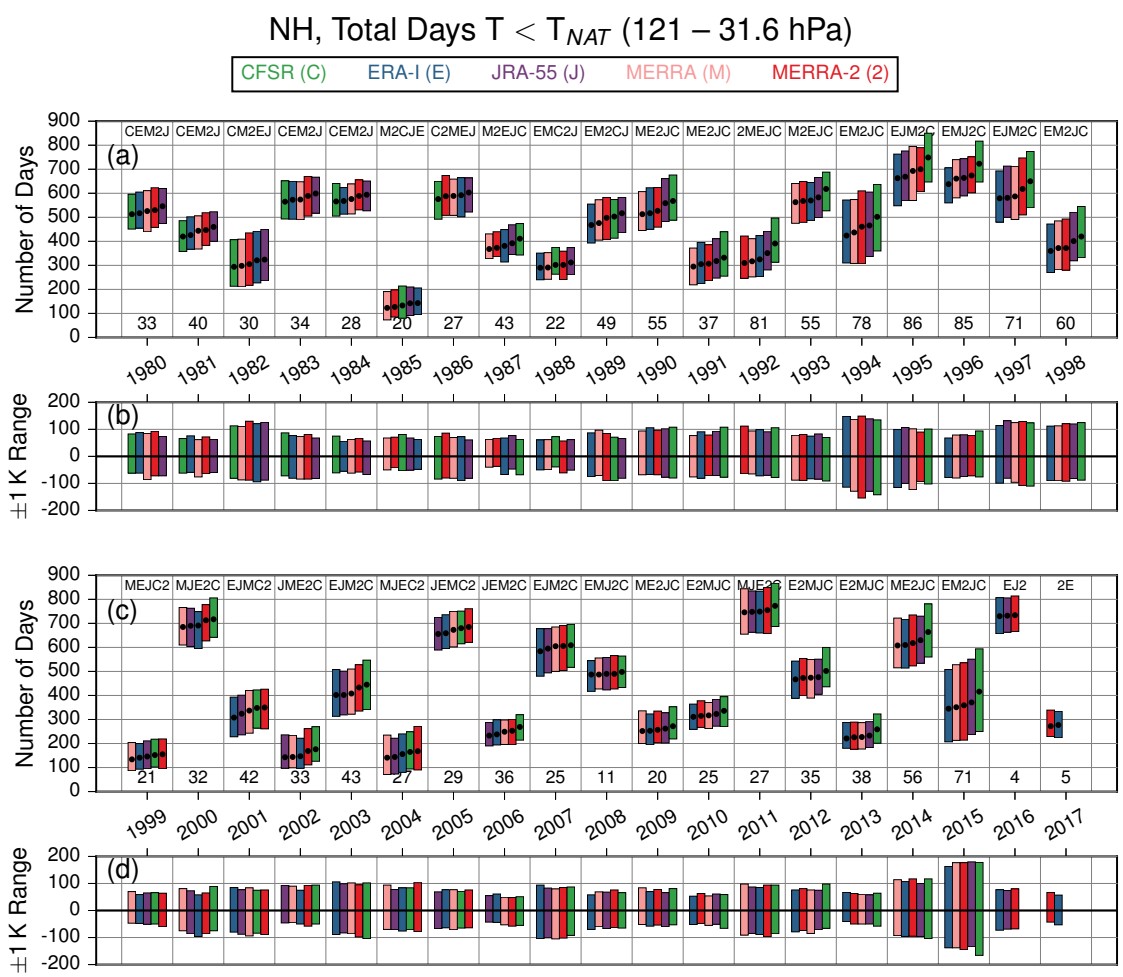

**Figure 17.** As in Figure 16 but for the NH and T$_{\text{NAT}}$. The days are counted from October through May.





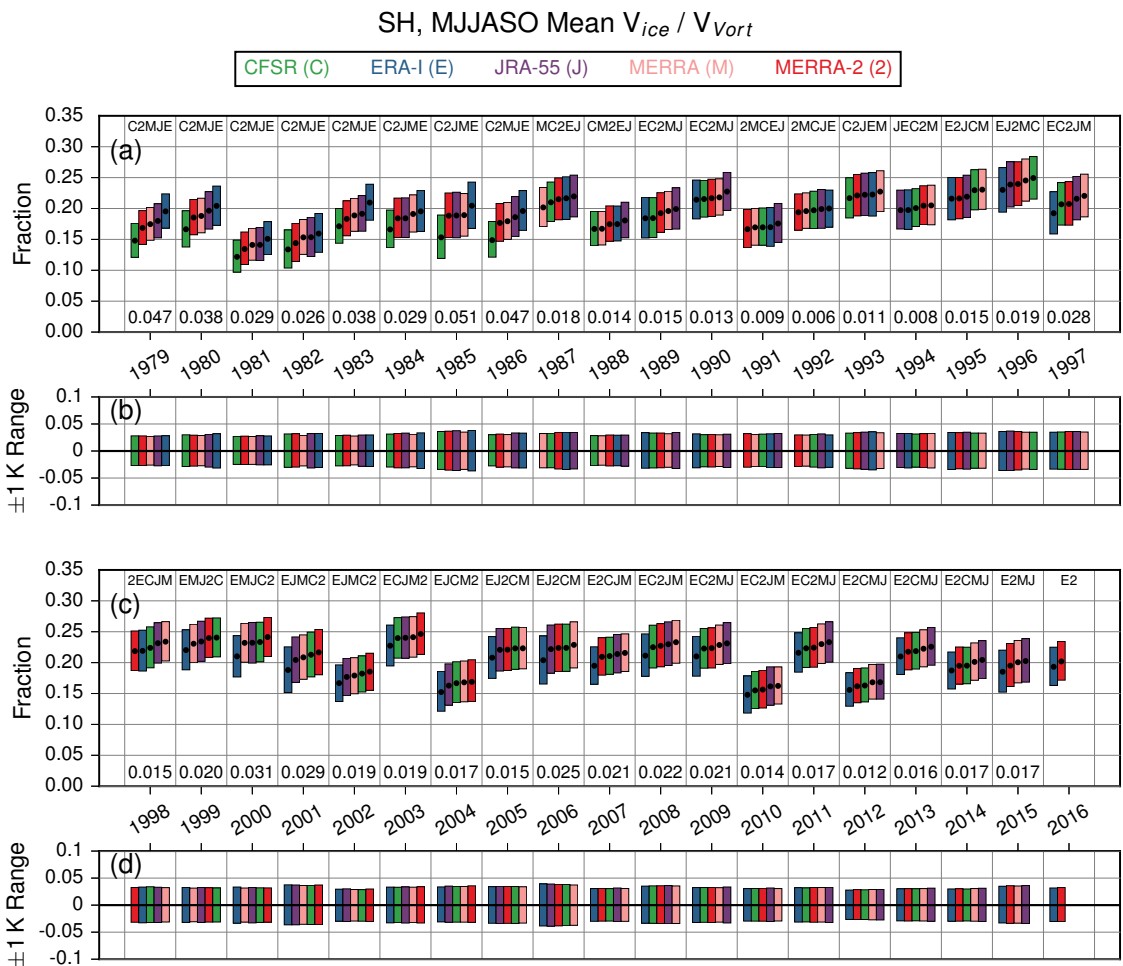

**Figure 18.** Winter mean fraction of vortex volume between the 390 and 580 K isentropic surfaces with temperatures below $T_{ice}$ in the SH (a and c), and range of values obtained for the $\pm 1$ K sensitivity tests. The winter mean is calculated over the full MJJASO period. The layout is as in Figure 16, with the numbers at the bottom of (a) and (c) being the range of central values (that is, rightmost minus leftmost central value). Green, blue, purple, pink, and red indicate CFSR, ERA-I, JRA-55, MERRA, and MERRA-2, respectively.





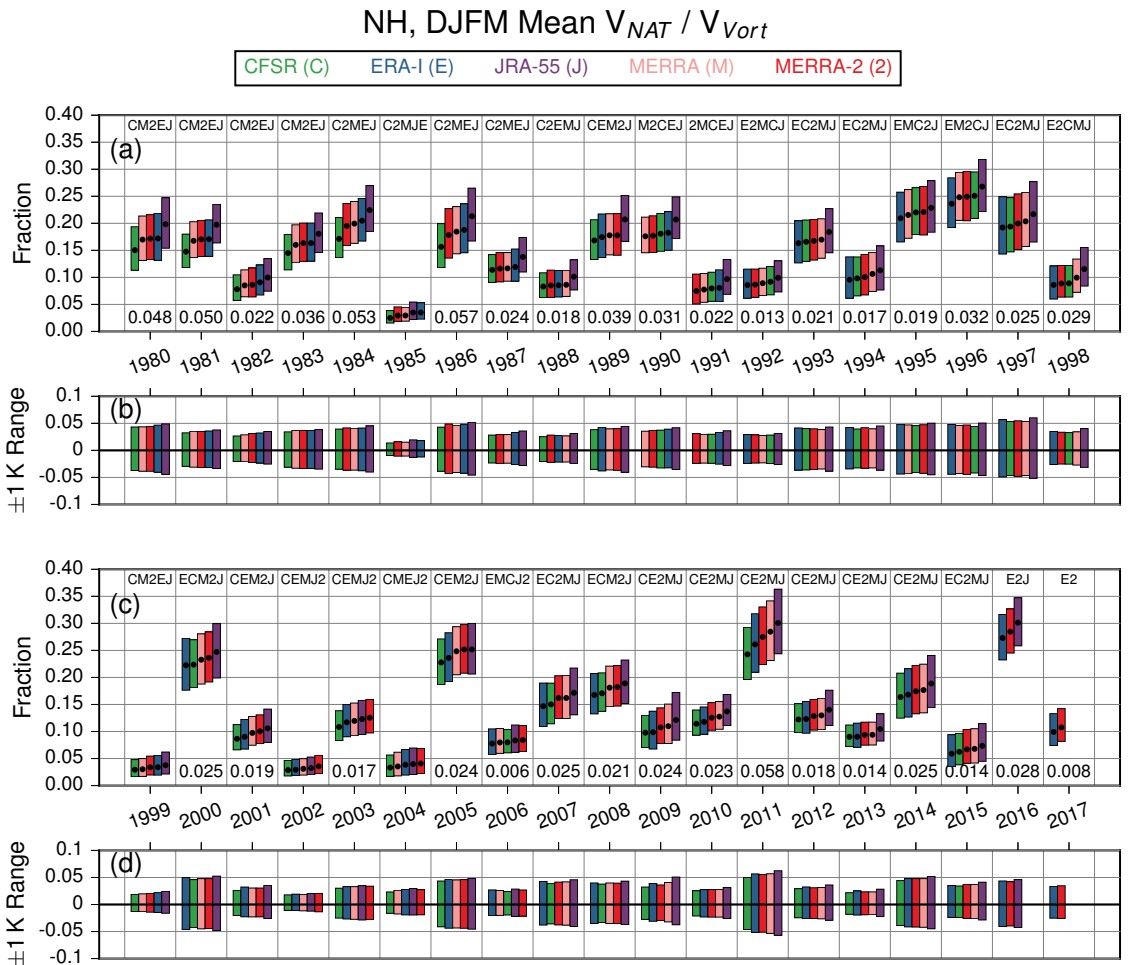

**Figure 19.** As in Figure 18 but for the NH and temperatures below $T_{NAT}$.





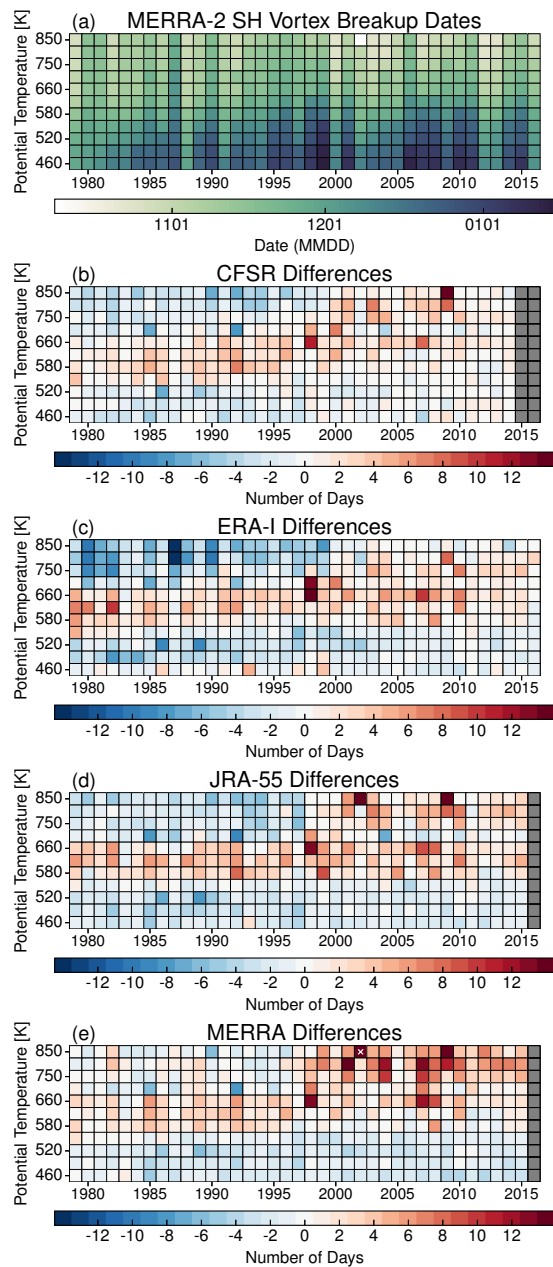

**Figure 20.** Pixel plots of (a) vortex decay dates (see text for the definition) in the MERRA-2 reanalysis, and (b through e) the difference between the vortex decay dates in each of the other reanalyses and MERRA-2 (as reanalysis − MERRA-2). The color bar ranges are restricted to distinguish differences of a few days; differences that greatly exceed the range by more than 7 days are marked with a white X.





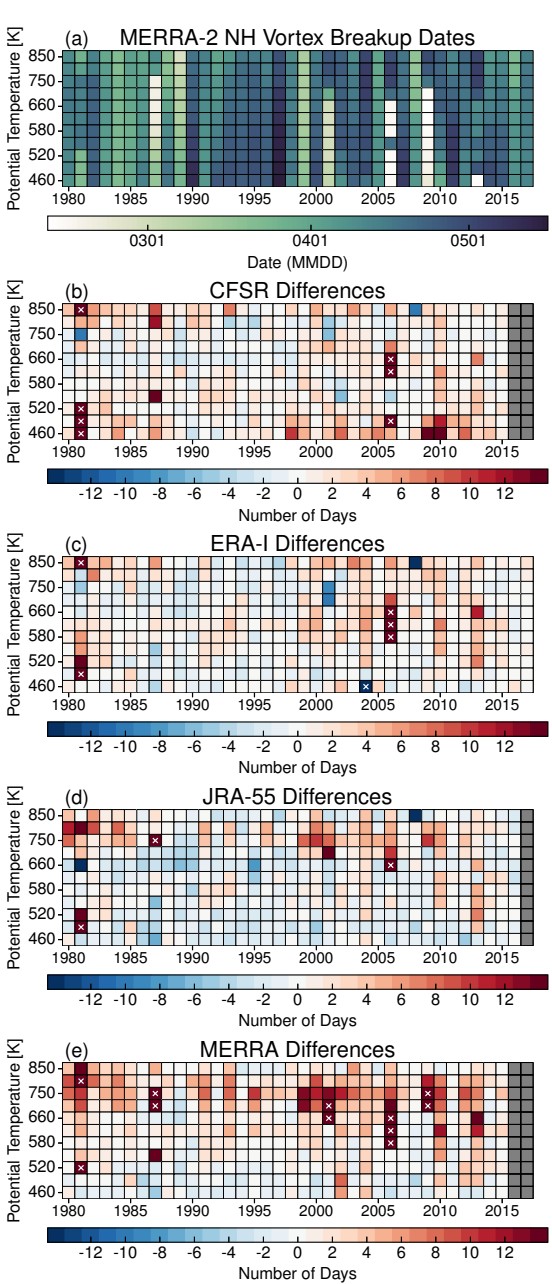

**Figure 21.** As in Figure 20 but for the NH.