# Peer review of "Reanalysis intercomparisons of stratospheric polar processing diagnostics"

_Atmospheric Chemistry and Physics, 2017_

## Referee Comment (RC1) · Anonymous Referee #2 · 21 Feb 2018

Reanalysis intercomparisons of stratospheric polar processing diagnostics Z. D. Lawrence, G. L. Manney, and K. Wargan

This manuscript provides an extensive intercomparison of diagnostics relevant for polar stratospheric ozone processing in five recent 'full-input' reanalyses, MERRA, MERRA2, CFSR, ERA Interim, and JRA55, as part of the S-RIP intercomparison project. The study is thorough, well thought out and generally clearly presented, and the intercomparison should provide a valuable reference point for studies of polar processing that are based on reanalysis data, as well as a reference point for comparisons of these quantities in future reanalyses. To me the more interesting results are the almost ubiquitous improvement seen in the agreement between reanalyses following the advent of improved satellite observations around 1998-2000, as well as the increased sensitivity to threshold definitions seen in the NH relative to the SH. The results are not earth shattering, but are of value and as such I would recommend publication after some minor revisions.

My main concern is that the paper is very long, and that its impact would be greater if it were significantly shorter. As a potential reference for future studies, there is some value in being rather complete in the intercomparisons, but 21 figures is a lot more than most readers will want to go through. It's not clear to me that Figs. 1-3 are really necessary, nor what is the additional gain from including Figs 18-19 over the content of Figs. 16-17.

Specific comments

p2 l1 There is a spurious 'data' here.

p2 l32: 'Best' is highly debatable here. They are a good tool, certainly, but they are not appropriate for all tasks.

p7 l6: The role of radiosondes should not be understated here – although it is not considered here, JRA55C, which assimilates only 'conventional' (non-satellite-based) observations does a remarkably good job of capturing much of the details of NH stratospheric variability.

p10 l4-19: The choice of a 5 day geometric mean here needs to be justified here. The key question is the decorrelation timescale of fluctuations in the differences between reanalyses. these could arise from a variety of processes with rather different timescales so it's not at all obvious to me what timescale is appropriate, but given that fluctuations in the physical quantities themselves (temperatures, PV) can have decorrelation timescales of far greater than 5 days this choice could be rendering the derived CIs rather meaningless. This can be checked directly by looking at the autocorrelation

functions of some sample quantities.

There is also a question of just what it means for two reanalyses to be 'statistically' indistinguishable. There is an important distinction to be drawn as to whether a difference seen between two temporal averages is indicative of a systematic, steady bias between the two systems as opposed to a result of the residual over temporal fluctuations. But given that these systems are meant to capture the same atmospheric fluctuations, time-dependent differences between reanalyses are still meaningful and potentially quite relevant to know about. Just because this measure indicates that the fluctuations are of larger amplitude than the mean bias (in some statistically meaningful sense) doesn't mean the reanalysis products are indistinguishable.

p10 l22-24: Are these averages and standard deviations taken over time (from the 12Z snapshots) within the year? Or are they taken over spatial degrees of freedom? Is the data synthetic? If not, what is actually shown?

A more general thought on this section - while I appreciate the effort to make the plots clear I wonder if it would be more efficient to simply explain this plot in the first case rather than present an example; the paper is quite long and omitting Figures 2 and 3 would go some ways towards shortening it without omitting relevant details.

Fig. 4: What is the relevance of the black lines 70 hPa and 30 hPa?

Fig. 5: Four digits of precision are not needed on the pressure axis labels

p13 l3: Earlier in the text A_PSC has been used - this to my mind is more standard than A_NAT. Was the switch intentional?

p15 l34: Up to 600K or so there is a significant improvement in the agreement between MERRA and MERRA 2 (in means and standard deviations) after 2000 - it's just in the upper stratosphere (particularly 660 and 700K) that the disagreement becomes if anything larger.

p16 l2: Is this a result of a more or less constant PV offset across the polar regions or

differences in the locations of the maximum gradient?

p16 l23: 'Total days' is a strange unit here since it's regularly far in excess of the total number of days in a year. The appropriate unit should be pressure-level days, I suppose.

p18 l26: I can't find an explicit definition of A_vort, though there are some relevant details in section 2.2.2

p22 l34: Given the statement two lines earlier about the similar timing of changes in the observations being assimilated by different reanalyses, the consistency of trends across multiple reanalyses should not be seen as any kind of definitive indication of the reliability of trends.

---

## Referee Comment (RC2) · S. Chabrillat (Referee) · 15 Apr 2018

**Referee comment on**
**"Reanalysis intercomparisons of stratospheric polar processing diagnostics" by Lawrence et al., ACPD, 2018.**

Simon Chabrillat (Royal Belgian Institute for Space Aeronomy, BIRA-IASB)

April 15, 2018

**1   General comments**

The authors provide a thorough and detailed intercomparison between five modern reanalyses of several diagnostics derived from polar lower stratospheric temperature and vorticity. This should go a long way towards assessing the reanalyses' representation of the potential for polar processing and ozone loss. Following the same approach as a previous paper comparing only ERA-I and MERRA, the focus is on the intercomparison between the reanalyses rather than a comparison with independent observations.

The text is well written with interesting and relevant information provided about the reanalyses and the outcome of the intercomparisons. The figures have been designed with great care to allow overviews of general tendencies as well as comparisons between the reanalyses on specific years. This should allow quick selections of winters with typical (or unusual) polar processing conditions while taking into account the agreement between the reanalyses (or lack thereof), which could be one of the major benefits of the paper. Unfortunately four basic questions are raised here about the methodology used for this comparison (see the major comments below, which are ordered by decreasing importance). It is possible that comments 2.2, 2.3 and 2.4 will be satisfactorily answered with a few sensitivity tests and deeper justifications. Yet I do not see how the choice of MERRA-2 as reference dataset can hold on close examination (for details see comment 2.1 below).

Indeed this manuscript shows that in the polar lower stratosphere MERRA-2 behaves quite differently from all other reanalyses before 1999. This result is important and should be highlighted in the abstract; several previous S-RIP papers reached similar conclusions in the tropical lower stratosphere. But it also invalidates the choice of MERRA-2 as a reference dataset to compute all means and standard deviations of the differences with the four other reanalyses. The choice of the average across four or five reanalyses would provide much additional information on the mean disagreements between the reanalyses and potentially on the variability of these disagreements. This additional information could very well change the conclusions of the study. Hence I believe that it is necessary to re-run at least all the difference diagnostics with another reference dataset and to update the methodology, discussion and conclusions accordingly.

**2 Major comments**

**2.1 Choice of MERRA-2 as a reference dataset**

Rather than using a Reanalysis Ensemble Mean (REM), the authors chose MERRA-2 as reference dataset to evaluate the differences between the reanalyses (shown on figs. 5 and 6, 8 and 9, 11 and 12, 14 and 15). In section 2.2.3 the rejection of a REM is first explained by the small size of the sample (5, or 4 if MERRA is excluded due to its expected similarity with MERRA-2) as this "can make the ensemble average sensitive to outliers". Yet this reasoning does not hold if the chosen reanalysis delivers itself many outlying diagnostics. Looking at the mean differences with CFSR, ERA-I and JRA-55 (and even MERRA in several cases; see left columns of figures listed above) the authors correctly note that before 1999 there are very similar band structures located in approximately the same layers and the standard deviations of the differences are remarkably similar in all four reanalyses. This indicates *a posteriori* that MERRA-2 is not an appropriate choice since it obscures the differences between the four other reanalyses.

In section 2.2.3 it is also explained that "comparing with an average across the reanalyses obscures the actual scale of differences among the individual datasets". Indeed it is quite desirable to show the spread across the reanalyses for each diagnostics, but this consideration should not influence the choice of the reference dataset. The spread can simply be shown by the difference between the maximum and the minimum value reached at each pixel for the considered diagnostic, or alternatively by the standard deviation of the 5 diagnostics (as done by Long et al., 2017, using only three reanalyses).

Since no REM was computed in this study, the authors are left with the difficult choice of one specific reanalysis as reference dataset. The justification for picking specifically MERRA-2 is only given in the conclusions: MERRA-2 is the most recent reanalysis (p.19, l.30). In the absence of independent observations it can be helpful to use such an arbitrary criterion but the selected reanalysis could turn out to be invalid as reference dataset due to special features (or even errors). Hence it is required to check a posteriori that the MERRA-2 diagnostics are inside the range provided by the other reanalyses. As discussed throughout the text this is precisely not the case - especially when one excludes MERRA which is an earlier version of the same reanalysis system and may have pre-processing issues (see next comment). Besides cursory examination of the figures listed above, many sentences in the manuscript highlight this fact (e.g. p.11, lines 32–33; p.12, lines 32–33; p.13, lines 28–29; p.14, line 24; p.15, lines 13–14 and 26; p.19 line 3) and the summary as well (section 4.1).

In the tropical regions, two earlier studies have shown that MERRA-2 does not represent correctly the Quasi-Biennal Oscillations before 1995 (Coy et al., 2016, doi:10.1175/JCLI-D-15-0809; Kawatani et al., 2016, doi:10.5194/acp-16-6681-2016). The present manuscript also mentions investigations in progress (Long et al., in preparation) showing that in the SH, the vertical profiles of temperature differences with sondes deliver vertical oscillations in one direction with ERA-I, in the other direction with MERRA-2 and no oscillations with the three other reanalyses (p.11-l.30 to p.12-l.4). These informations cast additional doubts on the pertinence of choosing MERRA-2 as reference dataset.

To summarize, I suggest to use the REM as reference dataset (also in the mean annual cycles in fig. 4, 7, 10, 13) and possibly to replace the standard deviations of the differences by one plot showing the spread of the diagnostics (or their standard deviations). The figures about inter-reanalysis (dis)agreements would show 5+1 plots instead of 4+4. Since the MERRA-2 diagnostics often fall outside the range found with other reanalyses, one could also try to highlight this with some simple line plots.

**2.2 Derivation of Potential Vorticity in the case of MERRA**

Many diagnostics are derived from temperature and Potential Vorticity (PV) on model levels and interpolated afterwards to isentropic levels. In the case of PV this raises special difficulties because only MERRA-2 provides it directly on model levels. Section 2.2.1 explains the preparation of this field in the four other reanalyses:

- ERA-I : PV is derived from absolute vorticity, $T$ and $p$ on model levels

- CFSR : PV is derived from relative vorticity, $T$ and $p$ on model levels

- JRA-55 : PV is derived from horizontal wind fields, $T$ and $p$ on model levels

- MERRA : PV is read on 42 pressure levels and interpolated back to model levels.

Hence the MERRA diagnostics on isentropic levels are the result of two successive vertical interpolations, which is not the case for any other reanalysis. These diagnostics are afterwards differentiated numerically w.r.t. equivalent latitude (for MPVG; see p. 9, lines 14–16) or determined from offset criteria (for Sunlit Vortex Averages, see for vortex decay dates, see p. 18, lines 25–28). The final quantities shown (i.e. differences between MERRA and MERRA-2 in figs 11 and 12, 14 and 15, 20 and 21) could turn out to be quite sensitive to this numerical issue.

Section 4.3 discusses this issue and correctly states that it is important to treat all reanalyses as fairly and equally as possible to reduce the uncertainty in sources of differences. As all five reanalyses, the MERRA dataset includes winds, $T$ and $p$ on model levels. Hence MERRA can easily be pre-processed in exactly the same manner as JRA-55, solving this issue once and for all (this approach could be applied to all five reanalyses to ensure strictly fair comparisons; yet in the experience of this reviewer, such pre-processing details are importantly mainly with respect to the vertical grid hence threaten only the results obtained with MERRA).

The fact that this issue is raised with MERRA is especially unfortunate because it is the reanalysis most similar to MERRA-2 which has itself been picked as reference dataset (see previous comment). There is a distinct possibility that a more consistent pre-processing would remove many differences between MERRA and MERRA-2. This would leave us with the expectable (hence unsatisfactory) conclusion that the reanalyses should be grouped between MERRA and MERRA-2 one one side and CFSR, JRA-55 and ERA-I on the other side.

**2.3 Analysis based only on daily fields valid at 12-UT**

Section 2.1 (p.5, lines 13–14) states that "*All analyses are done using daily 12-UT fields from each reanalysis dataset*". Are diurnal cycles completely negligible, even for diagnostics like $T_{min}$ when sunlight comes back? This seems like a serious assumption to me considering longitudinal asymmetries (look e.g. at Fig.5 in Lawrence et al., 2015) and also the real possibility that such diurnal cycles could be larger in some reanalyses than in other ones.

A simple way to check this assumption would be to re-run the diagnostics using e.g. only 0-UT fields. If the results turn out very similar, this sensitivity test would still be worth mentioning in the text. But any diagnostic showing non-negligible dependence on time of day (i.e. not the same results using 0-UT fields than 12-UT fields) should be run using 6-hourly fields as input.

**2.4 Definition of the vortex edge**

The discussion on Sunlit Vortex Area (p.16 lines 1–2) is not quite clear. I understand it as follows: "*Investigation of the reanalyses' differences in **total** vortex area  reveal**s** **that** they are nearly identical to the ones for SVA. This indicates that the **SVA** differences **from MERRA-2** are largely dominated by differences in  vortex  **area rather than vortex shape**.*". If this is correct, one wonders why the manuscript does not simply show, compare and discuss the vortex areas themselves.

Of course the determination of the vortex area closely depends on the method chosen to define the vortex edge. Here it is determined directly from the sPV values, using a constant vertical profile of sPV limits as a function of potential temperature (p. 9, lines 21–24 and Lawrence and Manney, 2017). This approach makes sense when using a single reanalysis, e.g. to interpret observations and their variations in time. Yet I wonder if this is valid when comparing several reanalyses because (contrarily to equivalent latitudes) they may have different ranges of sPV on any given day and isentropic level. It seems to me that the the classical vortex edge definition (equivalent latitude of the maximum of the wind speed times the PV gradient: Nash et al., 1996; Manney et al., JGR, 2007) could be more appropriate to the problem at hand because it would adapt itself to the different ranges of PV potentially delivered by each reanalysis. This could also deliver more robust evaluations of the vortex decay dates (p.19, lines 22–27).

**2.5 Need to add some auxiliary information**

Repeatability of these results requires several pieces of methodological information to be provided explicitly, e.g. in an annex or in a supplement:

- Last paragraph in section 2.2.1: please list of the 13 pressure levels used in the paper and their "*climatologically corresponding isentropic surfaces*".

- P.9, line 2: please provide the climatological profiles of $H_2O$ and $HNO_3$ used to determine $T_{ICE}$ and $T_{NAT}$ (with original reference if available).

- P.9, line 21–24: please list the sPV values defining the vortex edges at each isentropic level (unless of course the vortex edge definitions is changed - see previous comment).

**2.6 Opportunity to illustrate disagreements on a specific winter season**

One of the main goals of S-RIP is to increase in the community the awareness about the uncertainties hidden in the reanalyses, not only w.r.t. inter-annual variations but also for case studies. All diagnostics are shown as yearly time series which is useful to quickly evaluate the level of agreement between the reanalyses on any given winter. This provides an opportunity to highlight the potential disagreements between these diagnostics on a specific winter (and hemisphere) chosen for that purpose. So I suggest to select a diagnostic, year and level which highlight such disagreements between the reanalyses, and to plot this diagnostic either with line plots (e.g. time variations during that winter) or with maps (e.g. five maps for one specific date). Such an illustration would be easy to realize and could improve the impact of the paper.

For example, Figure 15 indicates that even on some recent years the difference of SVA between MERRA (or CFSR) and MERRA-2 can be as large as 2% of the NH, over vortex areas which have seasonal averages of around 8%. That seems quite large and may warrant a few detailed maps (unless of course this outcome does not hold after examination of the previous comments).

**3   Minor comments and corrections**

- Abstract: long enough that it would be useful to split it into two or three paragraphs.

- P.1, lines 16–17: *"Some reanalyses show convergence toward better agreement in vortex diagnostics after 1999, while others show some persistent differences across all years"* - please be specific, i.e. identify which reanalyses agree better and which ones have persistent differences.

- P.1, lines 24–25: *"the large interannual variability of NH winters has given rise to many seasons with marginal conditions and high sensitivity to reanalysis differences"*. Please re-phrase to be more precise: this sensitivity is clearly seen for the vortex decay dates (fig. 21) but not for the number of days (fig. 17) and fraction of vortex volumes (fig. 19) with $T < T_{NAT}$ .

- P.2, lines 21–22: or more correctly, *"detection of **recovery from chemical ozone depletion** also requires accurate knowledge of variability and long-term changes in polar vortex dynamics and temperatures."*

- P.2, line 23: *"the conversion of **chlorinated species** into forms..."*. Please also mention brominated species.

- P.2, line 29: Please define "active chlorine" (i.e. Cl and ClO)

- P.4, lines 5–6: *"...much larger temperature biases for NCEP/NCAR than in the other reanalyses, not only because that reanalysis is unsuitable for stratospheric studies..."* . This looks to me like a confusion between cause and effect. Consider writing instead something like *"...not only due to shortcomings of the former, but..."*

- P.4, line9: CFSR is not considered as a "full-input reanalysis"? Why?

- P.4, line 26: *"... during much of which ..."* - please re-phrase

- P.4, line 16–28: It seems to me that temperatures profiles retrieved from limb-scanning instruments (UARS-MLS, HALOE, SABER, MIPAS, Aura-MLS, ACE-FTS) could provide a valuable source of independent data to evaluate the reanalyses. Have such comparisons been done already for some reanalyses (or NWP analyses) with a focus on the polar lower stratosphere? If yes, the corresponding papers should be cited in the introduction. If no, this could mean that those instruments are not fit for this purpose and this warrants a short explanation in the introduction (e.g. due to lack of precision? lack of accuracy? lack of horizontal resolution?).

- P.4, lines 31–32: the history of the availability of specific reanalyses on native model levels is a quite technical matter for the introduction. I think that this sentence can be removed (such considerations are well explained in the next section).

- P.5, lines 14–17: Sentence is too long cumbersome (consider splitting). Replace *"...importance of resolution, especially the vertical grid,..."* with *"...importance of resolution, especially in the vertical dimension,..."*.

- P.5, line 19: define acronym "GMAO" or maybe drop it (anyway GMAO is the only division delivering atmospheric reanalyses at NASA).

- P.5, lines 23–24: I think that this approach is not valid, and that PV should be recomputed from $u, v, T, p$ on model levels as you did for JRA-55 (see major comment above).

- P.6, line 8: "...*but a much  **earlier** version than  in MERRA-2*".

- P.6, line 15: Please replace this URL by a proper bibliographic reference

- Section 2.1.2: the distinction between CFSR and CFSv2 is difficult to follow. The title of the section should be changed to the full name of the dataset, i.e. "CFSR/CFSv2". I advise to explain the distinction upfront: "*NCEP-CFSR (Saha et al., 2010) is a global reanalysis covering the period from 1979 to  2010. From 2011 onwards it is superseded by CFSv2 (Saha et al., 2014).*" and to finish the subsection with a simple sentence about the naming convention, e.g. "*Hereafter CFSR/CFSv2 is designated simply by CFSR*".

- Figure 1: Consider ending the caption with "*See Fujiwara (2017, Fig.8) for a similar time line but organized per instrument.*" (this is only a suggestion).

- Section 2.2.1: Consider citing also Manney et al. (JGR, 2007) which provided an excellent overview about the Derived Meteorological Products used here.

- Figure 2 and first paragraph of section 3: I do not think that these are really useful (they explain obvious concepts). Consider removing. If you decide to eep, line 23: replace "*To demonstrate...*" by "*To illustrate...*".

- P.11, lines 19–21: AIRS was introduced on September 2002 in MERRA and MERRA-25 and on February 2003 in ERA-I and CFSR. The potential importance of this instrument is an interesting point which deserves a few more details. For example, add a reference about its sensitivity to stratospheric temperatures. Similarly interesting comments could be added about GPS-RO which saw assimilation into ERA-I and CFSR from 2001, into MERRA-2 from 2004, into JRA-55 from 2006 and never into MERRA.

- P.12, line 30: "*...(as was the case in the SH) **they** remain larger in CFSR...*"

- P.12, line 35: "*...a clear decrease in them is not evident*". Please re-phrase.

- Discussion of figure 8 and 9 (p.13 line 26 to p.14 line 13): the standard deviations of the differences are not discussed at all, even though striking differences can be sen with the corresponding $T_{min}$ diagnostics in both hemispheres (i.e. compare right columns of fig.5 with fig.8 and fig.6 with fig.9). If you decide to keep showing standard deviations of differences (see first major comment) it would make sense to discuss this.

- P.14 line 29: remove words "*... show that the variances of the differences...*"

- P.15 line 6: see corresponding major comment (2.3)

- P.15 line 37: "*... (indicating **that** the**se** reanalyses hav **larger** SVA **than MERRA-2**)...*"

- Section 3.3 is very long and tedious to read, probably because it includes the methodology about the diagnostics shown here. Consider moving this methodology to a new subsection in section 2.

- Figures 16–17 and P.19 line 19: It looks like summing the number of days over lower stratospheric levels implies a close dependence on the vertical pressure grid used for this diagnostic. Is there a way to avoid this? In any case the explicit listing of these pressure levels is even more necessary (see major comment 2.4).

- Figures 16–19: I understand that plots (b) and (d) simply show the same sensitivity range as already shown by the bars in plots (a) and (d) but zoomed and centered on zero? If this is wrong, the captions and text require clarification. If this is right, the usefulness of plots (b) and (d) is not clear since they could be removed while not changing the discussion of the figures (also because the figures show sensitivities which do not depend much on the reanalyses nor on the year).

- P. 17 line 19: After 1999, it looks like fig. 17c has quasi-biennal periodicity. Could there be a link between this diagnostic and the QBO?

- P.17 line 21: Why is the important diagnostic $V_{PSC}/V_{vort}$ and not $V_{PSC}$ itself? If possible, explain this in one additional sentence.

- P.17 lines 27–28: "...*with a range* **among the reanalyses** *of 0.98 – 1.30 km...*"

- P.17 lines 28–29: I expect that the winter mean is applied at the end, i.e. you discuss winter means of daily fractions rather than the fractions of winter mean volumes? Please clarify, if possible in the caption of fig. 18 as well.

- P.18 lines 3–4: this last sentence is not useful (see also comment above about plots (b) and (d) not useful in these figures).

- P.18 lines 11–12: "...*with differences betweeen reanalyses indicating some differences in horizontal temperature gradients (especially in, e.g., 2011 and 2014).*". Can this be seen directly on fig. 19 or are you commenting figures which are not shown in the manuscript? Please clarify.

- Figures 20–21: please align the titles of the figures with the vocabulary used in the text (i.e. vortex decay dates - not vortex breakup dates). Since the ranges are the same for both figures, it is possible to clarify the caption: "...*differences*  **larger than 21 days** *are marked with a white X.*"

- P.18 line 35: "*For the SH, Figure 20a shows that...*"

- P.19 line 8: It is possible to get away from the figure and closer to its meaning. For example: "*Fig.20 also shows that on a few years (such as 2002 and 2009) the vortex decayed at a much later date in MERRA, JRA-55 and CFSR than in MERRA-2.*

- P.20 line 7: It is not fair to compare agreement **generally** within about 1K after 1998 with disagreements **up to** about 6K before.

- P.22 line 34: "...*and after rigorous assessment of the relationships of temperature changes to observations assimilated (which, to our knowledge has not been done).*" It could be that this has not yet been done systematically due to the huge diversity of assimilated observations. Yet Simmons et al. (QJRMS, 2014, doi:10.1002/qj.2317) already provided a remarkable first step in this direction.

- P.23, last sentence: "...*the comparison of reanalyses is a powerful tool for assessing robustness and uncertainty in these diagnostics.*" Yes but this is still an incomplete tool because the reanalyses often use similar parametrizations and assimilate very similar observational datasets.

---

## Author Comment (AC1) · 31 Jul 2018

**Responses to Reviewer 1's comments**

The reviewer's comments are given in *black italics* and our responses in blue plain text.

*This manuscript provides an extensive intercomparison of diagnostics relevant for polar stratospheric ozone processing in five recent 'full-input' reanalyses, MERRA, MERRA2, CFSR, ERA Interim, and JRA55, as part of the S-RIP intercomparison project. The study is thorough, well thought out and generally clearly presented, and the intercomparison should provide a valuable reference point for studies of polar processing that are based on reanalysis data, as well as a reference point for comparisons of these quantities in future reanalyses. To me the more interesting results are the almost ubiquitous improvement seen in the agreement between reanalyses following the advent of improved satellite observations around 1998-2000, as well as the increased sensitivity to threshold definitions seen in the NH relative to the SH. The results are not earth shattering, but are of value and as such I would recommend publication after some minor revisions.*

*My main concern is that the paper is very long, and that its impact would be greater if it were significantly shorter. As a potential reference for future studies, there is some value in being rather complete in the intercomparisons, but 21 figures is a lot more than most readers will want to go through. It's not clear to me that Figs. 1-3 are really necessary, nor what is the additional gain from including Figs 18-19 over the content of Figs. 16-17.*

We thank the reviewer for their helpful comments. The paper has been extensively revised in response to major comments by the other reviewer, Simon Chabrillat, so there is not a one-to-one correspondence with all of the specific suggestions made by the reviewer. We have, however, tried to keep specific new material as concise as possible and have removed material where it was suggested by either reviewer, as detailed in the specific responses below. This includes removing the original Figures 2, 3, 16, and 17 and the associated discussion.

Two major changes to the paper motivated by Simon Chabrillat's comments are to use a reanalysis ensemble mean (REM) as a reference for the comparisons rather than using MERRA-2 (see our response to Simon for discussion of this), and to remove MERRA from the reanalyses evaluated in this paper. There are numerous reasons for removing the MERRA comparisons, including the following: The choices that were made by GMAO of which products to archive for MERRA have made "fair" comparisons difficult to impossible for many products, including potential vorticity (PV), which is critical for stratospheric vortex and many other studies. While comparing MERRA with MERRA-2 and other reanalyses was critical to evaluating MERRA-2, numerous such studies have now been done; MERRA-2 was intended to supercede MERRA and sufficient evaluation of it has been done now to warrant this. Finally, especially when using the REM as a reference, it is somewhat problematic to include two reanalyses based on nearly the same model in a comparison of just five reanalyses.

Because these two major changes, especially the switch to using the REM, necessitated a nearly complete rewrite of large portions of the text in the results section (though the final results changed very little), several of the reviewers' comments now refer to text that has been replaced, and it is not possible to document every change in detail.

*Specific comments*
*p2 l1 There is a spurious 'data' here.*

Fixed.

*p2 l32: 'Best' is highly debatable here. They are a good tool, certainly, but they are not appropriate for all tasks.*

We have changed this to "among the best".

*p7 l6: The role of radiosondes should not be understated here – although it is not considered here, JRA55C, which assimilates only 'conventional' (non-satellite-based) observations does a remarkably good job of capturing much of the details of NH stratospheric variability.*

We have added a sentence noting the importance of radiosonde inputs in the lower stratosphere, but also noting the caveat that the sonde data are sparse in the NH polar regions and very sparse in the SH polar regions.

*p10 l4-19: The choice of a 5 day geometric mean here needs to be justified here. The key question is the decorrelation timescale of fluctuations in the differences between reanalyses. these could arise from a variety of processes with rather different timescales so it's not at all obvious to me what timescale is appropriate, but given that fluctuations in the physical quantities themselves (temperatures, PV) can have decorrelation timescales of far greater than 5 days this choice could be rendering the derived CIs rather meaningless. This can be checked directly by looking at the autocorrelation functions of some sample quantities.*

*There is also a question of just what it means for two reanalyses to be 'statistically' indistinguishable. There is an important distinction to be drawn as to whether a difference seen between two temporal averages is indicative of a systematic, steady bias between the two systems as opposed to a result of the residual over temporal fluctuations. But given that these systems are meant to capture the same atmospheric fluctuations, time-dependent differences between reanalyses are still meaningful and potentially quite relevant to know about. Just because this measure indicates that the fluctuations are of larger amplitude than the mean bias (in some statistically meaningful sense) doesn't mean the reanalysis products are indistinguishable.*

We have added justification for our choice of the expected block length for the stationary resampling procedure.

Since we moved to using a reanalysis ensemble mean (REM) based on Simon Chabrillat's review, we examined the autocorrelation functions (ACFs) for the differences of the reanalyses from the REM. What we found is that the decorrelation timescales can vary and depend highly on the reanalysis, the diagnostic, the year, and the vertical level; in some cases the decorrelation timescales reach zero in a few days, while in other cases they remain well above zero beyond 10 days. As examples of this, we have attached two figures of the type we used to evaluate these timescales at the end of our responses to reviewer 1. They are large and unwieldy figures, but they show (1) the ACF of the raw diagnostics for the REM (top panel) and the comparison reanalysis (second panel; in these cases MERRA-2), (2) the ACF of the comparison reanalysis minus REM (third panel), and (3) 18 ACFs of 18 different stationary resampled (with expected block length of 5 days) difference time series. The two examples we show here are for SH maximum PV gradients for the same level (490 K) separated by just one year. You can see that for 2015, the autocorrelation of the difference time series (3rd panel) stays fairly large out well beyond 10 days; in contrast, the ACF of the difference time series for 2014 drops much faster. You can also see that even though the decorrelation time scale is quite long for 2015 and the average block length of the resampled time series is 5 days, there are still a handful of resampled cases that also have relatively long decorrelation timescales (see e.g., n = 3, 9, 12, 16, 17, and 18) -- and there are also many resampled time series for 2014 that match the much shorter decorrelation time-scale pretty well too. This is one of the benefits of using the stationary resampling procedure rather than block resampling; using random block sizes can help to create artificial time series that better match the autocorrelation "structure" of the original time series.

After making and examining these sorts of plots, we repeated our bootstrapping procedure and tested using different expected block lengths between and including 5 and 15 days. What we found is that in all cases, the results we obtained were virtually identical. Ultimately, for the results now shown in the manuscript, we increased the expected block length to 10 days since it seemed to be the most "happy medium" among the many ACFs we examined; we also doubled the number of resamples for our bootstrap distributions to $2\times10^5$.

Regarding your second point, we agree that our results from the bootstrapping analysis should not be used to judge the (in)distinguishability of the reanalyses, but should be limited to the "classical" interpretation of statistical hypothesis testing. The presence of an "x" on our pixel plots (null hypothesis can't be rejected) does not mean that the time series of the certain diagnostic, year, and level are indistinguishable, just that we cannot reject that the winter means are equal. Conversely, the absence of an "x" on our pixel plots (null hypothesis rejected) does not mean that there are overwhelming or large biases, just that the winter means are unlikely to be equal. The significance testing here primarily supplements the winter mean differences and standard deviations -- for example, there are many cases of the diagnostic mean differences being very small but "significant" (no "x") alongside standard deviations that are very small,

which just says that although such differences are generally small, they are persistent enough during the season such that many resamples of the time series shared that persistent (but small) difference. There are also some cases where the diagnostic mean differences are noticeably nonzero but "insignificant" ("x" is present) alongside larger standard deviations, which indicates that the variability is large enough such that many resamples do not share the structures that give rise to the real mean difference.

We have modified and double checked our text to ensure we have not included any misleading language regarding the interpretation of the statistics.

*p10 l22-24: Are these averages and standard deviations taken over time (from the 12Z snapshots) within the year? Or are they taken over spatial degrees of freedom? Is the data synthetic? If not, what is actually shown?*

*A more general thought on this section - while I appreciate the effort to make the plots clear I wonder if it would be more efficient to simply explain this plot in the first case rather than present an example; the paper is quite long and omitting Figures 2 and 3 would go some ways towards shortening it without omitting relevant details.*

The data here were synthetic and meant to represent averages and standard deviations taken over time as in the other results we show, but we have taken your advice to shorten the paper and have ultimately taken out (what were formerly) Figures 2 and 3.

*Fig. 4: What is the relevance of the black lines 70 hPa and 30 hPa?*

These are the selected levels for which separate line plots are shown. This is now clarified in the caption.

*Fig. 5: Four digits of precision are not needed on the pressure axis labels*

The labels are now limited to a single digit after the decimal point in all the figures.

*p13 l3: Earlier in the text A_PSC has been used - this to my mind is more standard than A_NAT. Was the switch intentional?*

We use the subscripts _NAT and _ice to convey the particular type of PSC threshold we are looking at.  A note to this effect has been added in discussion of PSC thresholds in the methods section.

*p15 l34: Up to 600K or so there is a significant improvement in the agreement between MERRA and MERRA 2 (in means and standard deviations) after 2000 - it's just in the upper stratosphere (particularly 660 and 700K) that the disagreement becomes if anything larger.*

The MERRA comparisons have been removed from the paper for the reasons stated in our response to Simon Chabrillat, so this text has been removed.

*p16 l2: Is this a result of a more or less constant PV offset across the polar regions or differences in the locations of the maximum gradient?*

This text has been revised to reflect the individual calculations of the vortex edge location for each reanalysis. The results now suggest that this is related to differences in the locations of the maximum PV gradients, which is noted in the revised text.

*p16 l23: 'Total days' is a strange unit here since it's regularly far in excess of the total number of days in a year. The appropriate unit should be pressure-level days, I suppose.*

Because the V_PSC / V_vort figures provide much of the same information, and to shorten the paper, we have followed your suggestion to delete the plots showing days integrated over the levels, so these figures have been removed.

*p18 l26: I can't find an explicit definition of A_vort, though there are some relevant details in section 2.2.2*

Since we do not use "A_vort" elsewhere in the paper, we now simply refer to it as "vortex area". We have also made the definition of vortex area more explicit. However, please note that the paragraphs discussing the methods behind the derived diagnostics have been moved to a new subsubsection of section 2 (in response to a comment by Simon Chabrillat).

*p22 l34: Given the statement two lines earlier about the similar timing of changes in the observations being assimilated by different reanalyses, the consistency of trends across multiple reanalyses should not be seen as any kind of definitive indication of the reliability of trends.*

Agreement across reanalyses would be a ***necessary*** condition to believe trends derived from them to be reliable. We agree that it is certainly not a ***sufficient*** condition. We have reworded the sentence in question to make this more explicit.

---

## Author Comment (AC2) · 31 Jul 2018

**Responses to Simon Chabrillat's Comments**

Simon's comments are given in *black italics* and our responses in blue.

*General comments*

*The authors provide a thorough and detailed intercomparison between five modern reanalyses of several diagnostics derived from polar lower stratospheric temperature and vorticity. This should go a long way towards assessing the reanalyses' representation of the potential for polar processing and ozone loss. Following the same approach as a previous paper comparing only ERA-I and MERRA, the focus is on the intercomparison between the reanalyses rather than a comparison with independent observations.*

*The text is well written with interesting and relevant information provided about the reanalyses and the outcome of the intercomparisons. The figures have been designed with great care to allow overviews of general tendencies as well as comparisons between the reanalyses on specific years. This should allow quick selections of winters with typical (or unusual) polar processing conditions while taking into account the agreement between the reanalyses (or lack thereof), which could be one of the major benefits of the paper. Unfortunately four basic questions are raised here about the methodology used for this comparison (see the major comments below, which are ordered by decreasing importance). It is possible that comments 2.2, 2.3 and 2.4 will be satisfactorily answered with a few sensitivity tests and deeper justifications. Yet I do not see how the choice of MERRA-2 as reference dataset can hold on close examination (for details see comment 2.1 below).*

*Indeed this manuscript shows that in the polar lower stratosphere MERRA-2 behaves quite differently from all other reanalyses before 1999. This result is important and should be highlighted in the abstract; several previous S-RIP papers reached similar conclusions in the tropical lower stratosphere. But it also invalidates the choice of MERRA-2 as a reference dataset to compute all means and standard deviations of the differences with the four other reanalyses. The choice of the average across four or five reanalyses would provide much additional information on the mean disagreements between the reanalyses and potentially on the variability of these disagreements. This additional information could very well change the conclusions of the study. Hence I believe that it is necessary to re-run at least all the difference diagnostics with another reference dataset and to update the methodology, discussion and conclusions accordingly.*

We thank Simon for his thoughtful and detailed comments.  Two major changes to the paper motivated by his comments are to use a reanalysis ensemble mean (REM) as a reference for the comparisons rather than using MERRA-2 (see our detailed response below), and to remove MERRA from the reanalyses evaluated in this paper.  There are numerous reasons for removing the MERRA comparisons, including the following:  The choices that were made by GMAO of which products to archive for MERRA have made "fair" comparisons difficult to impossible for many products, including potential vorticity (PV), which is critical for stratospheric vortex and many other studies.  While comparing MERRA with MERRA-2 and other reanalyses was critical to evaluating MERRA-2, numerous such studies have now been done; MERRA-2 was intended

to supercede MERRA and sufficient evaluation of it has been done now to warrant this. Finally, especially when using the REM as a reference, it is somewhat problematic to include two reanalyses based on nearly the same model in a comparison of just five reanalyses.

Because these two major changes, especially the switch to using the REM, necessitated a nearly complete rewrite of large portions of the text in the results section (though the final conclusions and relationships among the reanalyses don't change), several of the reviewers' comments now refer to text that has been replaced, and it is not possible to document every change in detail.

*Major comments*
*2.1*
*Choice of MERRA-2 as a reference dataset*
*Rather than using a Reanalysis Ensemble Mean (REM), the authors chose MERRA-2 as reference dataset to evaluate the differences between the reanalyses (shown on figs. 5 and 6, 8 and 9, 11 and 12, 14 and 15). In section 2.2.3 the rejection of a REM is first explained by the small size of the sample (5, or 4 if MERRA is excluded due to its expected similarity with MERRA-2) as this "can make the ensemble average sensitive to outliers". Yet this reasoning does not hold if the chosen reanalysis delivers itself many outlying diagnostics. Looking at the mean differences with CFSR, ERA-I and JRA-55 (and even MERRA in several cases; see left columns of figures listed above) the authors correctly note that before 1999 there are very similar band structures located in approximately the same layers and the standard deviations of the differences are remarkably similar in all four reanalyses. This indicates a posteriori that MERRA-2 is not an appropriate choice since it obscures the differences between the four other reanalyses.*

*In section 2.2.3 it is also explained that "comparing with an average across the reanalyses obscures the actual scale of differences among the individual datasets". Indeed it is quite desirable to show the spread across the reanalyses for each diagnostics, but this consideration should not influence the choice of the reference dataset. The spread can simply be shown by the difference between the maximum and the minimum value reached at each pixel for the considered diagnostic, or alternatively by the standard deviation of the 5 diagnostics (as done by Long et al., 2017, using only three reanalyses).*

*Since no REM was computed in this study, the authors are left with the difficult choice of one specific reanalysis as reference dataset. The justification for picking specifically MERRA-2 is only given in the conclusions: MERRA-2 is the most recent reanalysis (p.19, l.30). In the absence of independent observations it can be helpful to use such an arbitrary criterion but the selected reanalysis could turn out to be invalid as reference dataset due to special features (or even errors). Hence it is required to check a posteriori that the MERRA-2 diagnostics are inside the range provided by the other reanalyses. As discussed throughout the text this is precisely not the case - especially when one excludes MERRA which is an earlier version of the same reanalysis system and may have pre-processing issues (see next comment). Besides cursory examination of the figures listed above, many sentences in the manuscript highlight this fact (e.g. p.11, lines 32–33; p.12, lines 32–33; p.13, lines 28–29; p.14, line 24; p.15, lines 13–14*

*and 26; p.19 line 3) and the summary as well (section 4.1).*

*In the tropical regions, two earlier studies have shown that MERRA-2 does not represent correctly the Quasi-Biennal Oscillations before 1995 (Coy et al., 2016, doi:10.1175/JCLI-D-15-0809; Kawatani et al., 2016, doi:10.5194/acp-16-6681-2016). The present manuscript also mentions investigations in progress (Long et al., in preparation) showing that in the SH, the vertical profiles of temperature differences with sondes deliver vertical oscillations in one direction with ERA-I, in the other direction with MERRA-2 and no oscillations with the three other reanalyses (p.11-l.30 to p.12-l.4). These informations cast additional doubts on the pertinence of choosing MERRA-2 as reference dataset.*

*To summarize, I suggest to use the REM as reference dataset (also in the mean annual cycles in fig. 4, 7, 10, 13) and possibly to replace the standard deviations of the differences by one plot showing the spread of the diagnostics (or their standard deviations). The figures about inter-reanalysis (dis)agreements would show 5+1 plots instead of 4+4. Since the MERRA-2 diagnostics often fall outside the range found with other reanalyses, one could also try to highlight this with some simple line plots.*

We have switched to using the REM as the reference dataset, but we kept our "pixel plots" in the same format as before. We feel that our change to using the REM is overall beneficial, because while our results did not change much, this change demonstrated that none of the reanalyses are outliers (to be fair, we were also guilty of using this language when we said that the REM could be sensitive to outliers). In a sample size of 4 to 5, one cannot make judgments about the suitability of a reanalysis as a reference unless there is evidence of it being egregiously different, and using the REM here has shown that to not be the case for any of the reanalyses.

*2.2*
*Derivation of Potential Vorticity in the case of MERRA*

*Many diagnostics are derived from temperature and Potential Vorticity (PV) on model levels and interpolated afterwards to isentropic levels. In the case of PV this raises special difficulties because only MERRA-2 provides it directly on model levels. Section 2.2.1 explains the preparation of this field in the four other reanalyses:*
*• ERA-I : PV is derived from absolute vorticity, T and p on model levels*
*• CFSR : PV is derived from relative vorticity, T and p on model levels*
*• JRA-55 : PV is derived from horizontal wind fields, T and p on model levels*
*• MERRA : PV is read on 42 pressure levels and interpolated back to model levels.*
*Hence the MERRA diagnostics on isentropic levels are the result of two successive vertical interpolations, which is not the case for any other reanalysis. These diagnostics are afterwards differentiated numerically w.r.t. equivalent latitude (for MPVG; see p. 9, lines 14–16) or determined from offset criteria (for Sunlit Vortex Averages, see for vortex decay dates, see p. 18, lines 25–28). The final quantities shown (i.e. differences between MERRA and MERRA-2 in figs 11 and 12, 14 and 15, 20 and 21) could turn out to be quite sensitive to this numerical Issue.*

*Section 4.3 discusses this issue and correctly states that it is important to treat all reanalyses as fairly and equally as possible to reduce the uncertainty in sources of differences. As all five reanalyses, the MERRA dataset includes winds, T and p on model levels. Hence MERRA can easily be pre-processed in exactly the same manner as JRA-55, solving this issue once and for all (this approach could be applied to all five reanalyses to ensure strictly fair comparisons; yet in the experience of this reviewer, such pre-processing details are importantly mainly with respect to the vertical grid hence threaten only the results obtained with MERRA). The fact that this issue is raised with MERRA is especially unfortunate because it is the reanalysis most similar to MERRA-2 which has itself been picked as reference dataset (see previous comment). There is a distinct possibility that a more consistent pre-processing would remove many differences between MERRA and MERRA-2. This would leave us with the expectable (hence unsatisfactory) conclusion that the reanalyses should be grouped between MERRA and MERRA-2 one one side and CFSR, JRA-55 and ERA-I on the other side.*

The unavailability of PV or vorticity on the model grid in MERRA has been a hindrance to scientific and comparison studies since its production. In addition to the different calculations Simon mentions here, the PV that is available from MERRA is from the 'ANA' rather than the 'ASM' fields, but the latter are the recommended ones for most purposes (e.g., Fujiwara et al., 2017). The is one of several reasons (see above for others) that we have chosen to remove MERRA from the reanalyses evaluated in this paper.

Even without MERRA, the non-uniformity in PV fields available from different reanalyses is a concern in numerous studies, as we have discussed in the implications and recommendations in our conclusions. A detailed study of differences in PV fields, and the effects of calculating them differently, is obviously beyond the scope of this paper (though it would be valuable and we are initiating such a study). However: (1) the differences in PV fields are most important here for the diagnostics that rely on identifying the vortex edge, and we have made this process more uniform for all the reanalyses as detailed in our response to Simon's major comment 2.4 below; and (2) in response to Simon's last point above, along with other diagnostics, when compared to the REM, it is apparent that MERRA-2 (and MERRA, though no longer shown in the paper) is not a consistent outlier in these diagnostics.

*2.3*
 *Analysis based only on daily fields valid at 12-UT*
*Section 2.1 (p.5, lines 13–14) states that "All analyses are done using daily 12-UT fields from each reanalysis dataset". Are diurnal cycles completely negligible, even for diagnostics like Tmin when sunlight comes back? This seems like a serious assumption to me considering longitudinal asymmetries (look e.g. at Fig.5 in Lawrence et al., 2015) and also the real possibility that such diurnal cycles could be larger in some reanalyses than in other ones.*
*A simple way to check this assumption would be to re-run the diagnostics using e.g. only 0-UT fields. If the results turn out very similar, this sensitivity test would still be worth mentioning in the text. But any diagnostic showing non-negligible dependence on time of day (i.e. not the same results using 0-UT fields than 12-UT fields) should be run using 6-hourly*

*fields as input.*

We re-ran all of our diagnostics using 00UT, and repeated our analyses -- everything came out virtually identical. We now mention in the paper that we have tested using 00UT data and that it does not affect our results.

*2.4*

 *Definition of the vortex edge*

*The discussion on Sunlit Vortex Area (p.16 lines 1–2) is not quite clear. I understand it as*

*follows: "Investigation of the reanalyses' differences in total vortex area from MERRA-2 reveals that they are nearly identical to the ones for SVA. This indicates that the SVA differences from MERRA-2 are largely dominated by differences in the size of the vortex edge contours area rather than vortex shape.". If this is correct, one wonders why the manuscript does not simply show, compare and discuss the vortex areas themselves.*

*Of course the determination of the vortex area closely depends on the method chosen to define the vortex edge. Here it is determined directly from the sPV values, using a constant vertical profile of sPV limits as a function of potential temperature (p. 9, lines 21–24 and Lawrence and Manney, 2017). This approach makes sense when using a single reanalysis, e.g. to interpret observations and their variations in time. Yet I wonder if this is valid when comparing several reanalyses because (contrarily to equivalent latitudes) they may have different ranges of sPV on any given day and isentropic level. It seems to me that the the classical vortex edge definition (equivalent latitude of the maximum of the wind speed times the PV gradient: Nash et al., 1996; Manney et al., JGR, 2007) could be more appropriate to the problem at hand because it would adapt itself to the different ranges of PV potentially delivered by each reanalysis. This could also deliver more robust evaluations of the vortex decay dates (p.19, lines 22–27).*

The Nash method and the windspeed*PV gradient methods are slightly different (see, e.g., Manney et al., 2007, JGR for a discussion of this), but the main issue is that using any method to get daily varying vortex edge values would complicate and possibly contaminate the intercomparisons. Daily methods are prone to giving spurious jumps and oscillations that can dramatically change the vortex edge value (and thus quantities such as vortex area) from one day to the next. If this happened at different times in different reanalyses, the results could be unnecessarily skewed. As examples of this behavior, consider the vortex area and vortex edge quantities provided on the ozonewatch.gsfc.nasa.gov website, which catalogs these quantities for MERRA-2 data (which we have confirmed are based on the Nash method).

The following is vortex area (in million km^2) and vortex edge PV (as "modified" PV) for the currently ongoing SH winter at 460K:

| Date | Area (km^2) | VortEdge MPV |
|------|-------------|--------------|
| 2018-07-09 | 18.90 | -27.15 |
| 2018-07-10 | 23.76 | -25.00 |
| 2018-07-11 | 29.05 | -22.80 |
| 2018-07-12 | 35.04 | -20.26 |
| 2018-07-13 | 39.77 | -18.30 |
| 2018-07-14 | 45.01 | -16.17 |

In this case, the vortex apparently grows nearly 5 million square kilometers per day (roughly 2% of a hemisphere per day), from an all time climatological minimum vortex area to above 90th percentile (for this level), all in the span of 5 days.  Examination of PV maps does not support there being a significant change in the vortex area at this time
[see the plot at
https://ozonewatch.gsfc.nasa.gov/meteorology/figures/merra2/pv/mpvweas_460_2018_merra2.pdf and the data from
https://ozonewatch.gsfc.nasa.gov/meteorology/figures/merra2/pv/mpvweas_460_2018_merra2.pdf and
https://ozonewatch.gsfc.nasa.gov/meteorology/figures/merra2/pv/mpvwes_460_2018_merra2.txt;
PV maps can be viewed at https://acd-ext.gsfc.nasa.gov/Data_services/antarctic/history.html].

Note that cases such as this one are not unique; other examples in other years, at other levels, and in both hemispheres are relatively common. For these reasons (also see the discussion in Manney et al., 2007, JGR, regarding the robustness of daily varying versus constant vortex edge values), we do not like to use daily varying vortex edges and prefer instead to use constant vortex edges. Although using constant vortex edges is also unrealistic in the sense that a single PV value on a single isentropic level cannot be the vortex edge forever, as long as an appropriate value is chosen within the strong PV gradient region near the vortex edge, the chosen contour will grow and decay in a realistic and more gradual manner that is still representative (at some times more so than a time-varying value) of the vortex.

We have switched to using constant vortex edge values that were calculated for each reanalysis individually, using the climatological average PV values at the maximum PV gradients. We chose to strictly use PV to determine the edges in this case so that we would not introduce another field/quantity (winds or otherwise), with its own uncertainties, into the mix.

*2.5*
*Need to add some auxiliary information*
*Repeatability of these results requires several pieces of methodological information to be provided explicitly, e.g. in an annex or in a supplement:*
*• Last paragraph in section 2.2.1: please list of the 13 pressure levels used in the paper and their "climatologically corresponding isentropic surfaces".*

We now list the pressures and potential temperatures of the levels (there are actually 14 including both "boundary" ones) that are used in the appendix.

• P.9, line 2: please provide the climatological profiles of H2 O and HNO3 used to determine TICE and TN AT (with original reference if available).

A table of the values and paragraph on how they are calculated have been added as an appendix.

• P.9, line 21–24: please list the sPV values defining the vortex edges at each isentropic level (unless of course the vortex edge definitions is changed - see previous comment).

Because we now calculate the vortex edge sPV values used as a function of altitude for each reanalysis separately (as described in response to 2.4), a new figure has been added that shows the profile of vortex edge sPV values used for each reanalysis. As noted in the "data availability" text, our diagnostics (including the vortex edge profiles used) are available by contacting the authors.

*2.6*
 *Opportunity to illustrate disagreements on a specific winter season*
*One of the main goals of S-RIP is to increase in the community the awareness about the uncertainties hidden in the reanalyses, not only w.r.t. inter-annual variations but also for case studies. All diagnostics are shown as yearly time series which is useful to quickly evaluate the level of agreement between the reanalyses on any given winter. This provides an opportunity to highlight the potential disagreements between these diagnostics on a specific winter (and hemisphere) chosen for that purpose. So I suggest to select a diagnostic, year and level which highlight such disagreements between the reanalyses, and to plot this diagnostic either with line plots (e.g. time variations during that winter) or with maps (e.g. five maps for one specific date). Such an illustration would be easy to realize and could improve the impact of the paper. For example, Figure 15 indicates that even on some recent years the difference of SVA between MERRA (or CFSR) and MERRA-2 can be as large as 2% of the NH, over vortex areas which have seasonal averages of around 8%. That seems quite large and may warrant a few detailed maps (unless of course this outcome does not hold after examination of the previous comments).*

While an examination of differences in case studies among the reanalyses would be an interesting and potentially valuable study, examining just one case would provide an incomplete and possibly misleading impression of the detailed reasons for and morphology underlying the differences we show. The differences and reasons for them will vary by hemisphere, by the time period chosen because of different data inputs into the reanalyses, and, especially in the NH, because of interannual and intraseasonal variability in meteorological conditions (e.g., for disturbed versus quiescent meteorological conditions). Because this type of analysis would be

worth a more detailed study on its own where sensitivity to such conditions could be explored, and because our paper was already too long, we consider this beyond the scope of this paper.

 *Minor comments and corrections*
*• Abstract: long enough that it would be useful to split it into two or three paragraphs.*

We have divided the abstract into several paragraphs (after modifying it to reflect the changes in the paper) to more clearly separate different results highlighted.

*• P.1, lines 16–17: "Some reanalyses show convergence toward better agreement in vortex diagnostics after 1999, while others show some persistent differences across all years" - please be specific, i.e. identify which reanalyses agree better and which ones have persistent differences.*

This text has been modified to reflect the patterns seen in differences from the REM, so the text in question has been removed.

*• P.1, lines 24–25: "the large interannual variability of NH winters has given rise to many seasons with marginal conditions and high sensitivity to reanalysis differences". Please re-phrase to be more precise: this sensitivity is clearly seen for the vortex decay dates (fig. 21) but not for the number of days (fig. 17) and fraction of vortex volumes (fig. 19) with T < TN AT.*

We have reworded this sentence to indicate that the results are more sensitive in the NH and that this is particularly apparent in the vortex decay dates.

*• P.2, lines 21–22: or more correctly, "detection of recovery from chemical ozone depletion also requires accurate knowledge of variability and long-term changes in polar vortex dynamics and temperatures."*

We have changed this as suggested.

*• P.2, line 23: "the conversion of chlorinated species into forms...". Please also mention brominated species.*

We have changed the wording as suggested and added "and brominated".

*• P.2, line 29: Please define "active chlorine" (i.e. Cl and ClO)*

We now spell it out: Cl+ClO+2ClOOCl

*• P.4, lines 5–6: "...much larger temperature biases for NCEP/NCAR than in the other*

*reanalyses, not only because that reanalysis is unsuitable for stratospheric studies...".
This looks to me like a confusion between cause and effect. Consider writing instead
something like "...not only due to shortcomings of the former, but..."*

We have reworded this along the lines suggested.

• *P.4, line9: CFSR is not considered as a "full-input reanalysis"? Why?*

This mention of "full-input reanalysis" was within a parenthetical where we were explaining
JRA-55 as "the Japan Meteorological Agency's latest reanalysis assimilating both surface and
upper air observations"; we used this same parenthetical to explain that a reanalysis that
assimilates both surface and upper air observations is considered full-input. We did not intend
for it to sound as if CFSR was not a full-input reanalysis. While we did/do refer to CFSR/CFSv2
as a full-input reanalysis elsewhere, we understand that this text is confusing, so we have
moved the full-input description elsewhere to be more clear.

• *P.4, line 26: "... during much of which ..." - please re-phrase*

We rephrased this sentence.

• *P.4, line 16–28: It seems to me that temperatures profiles retrieved from limb-scanning in-
struments (UARS-MLS, HALOE, SABER, MIPAS, Aura-MLS, ACE-FTS) could provide
a valuable source of independent data to evaluate the reanalyses. Have such comparisons
been done already for some reanalyses (or NWP analyses) with a focus on the polar lower
stratosphere? If yes, the corresponding papers should be cited in the introduction. If no,
this could mean that those instruments are not fit for this purpose and this warrants a
short explanation in the introduction (e.g. due to lack of precision? lack of accuracy?
lack of horizontal resolution?).*

The introduction already had two paragraphs describing previous studies comparing polar
processing diagnostics with observational data (in the discussion paper, page 3 line 32 through
page 4 line 28).  In addition, part of one of those paragraphs discusses why the diagnostics we
compare here cannot be easily compared with observations (in the discussion paper, page 4,
lines 21 through 26).  We have, however, expanded a bit in the first of those paragraphs on why
it is difficult (sometimes impossible) to compare polar processing diagnostics with limb-sounding
data.

• *P.4, lines 31–32: the history of the availability of specific reanalyses on native model levels
is a quite technical matter for the introduction. I think that this sentence can be removed
(such considerations are well explained in the next section).*

We have deleted this sentence as suggested.

*• P.5, lines 14–17: Sentence is too long cumbersome (consider splitting). Replace "...importance of resolution, especially the vertical grid,..." with "...importance of resolution, especially in the vertical dimension,...".*

We replaced "the vertical grid" with "in the vertical dimension" as you suggest. As we no longer include MERRA in the mix the "except where unavailable (e.g., PV for MERRA)" bit got deleted. We think and hope the sentence reads fine now.

*• P.5, line 19: define acronym "GMAO" or maybe drop it (anyway GMAO is the only division delivering atmospheric reanalyses at NASA).*

GMAO is now defined.

*• P.5, lines 23–24: I think that this approach is not valid, and that PV should be recomputed from u, v, T, p on model levels as you did for JRA-55 (see major comment above).*

As noted in the major changes and the response to major comment 2.2, we have removed MERRA from the intercomparison, so this text has been removed.

*• P.6, line 8: "...but a much older earlier version than that used in MERRA-2 ".*

The text read: "a much older version". This sentence was deleted because we no longer analyze MERRA.

*• P.6, line 15: Please replace this URL by a proper bibliographic reference*

We have reformatted the citation to give the URL in a proper bibliographic reference.

*• Section 2.1.2: the distinction between CFSR and CFSv2 is difficult to follow. The title of the section should be changed to the full name of the dataset, i.e. "CFSR/CFSv2".*
*I advise to explain the distinction upfront: "NCEP-CFSR/CFSv2 (hereinafter CFSR) (Saha et al., 2010) is a global reanalysis covering the period from 1979 to the present 2010. From 2011 onwards it is superseded by CFSv2 (Saha et al., 2014)."*
*and to finish the subsection with a simple sentence about the naming convention, e.g. "Hereafter CFSR/CFSv2 is designated simply by CFSR".*

Consistent with recently clarified recommendations from the S-RIP project, we now use "CFSR/CFSv2" throughout the paper.  We have clarified the text regarding the transition between the two.

*• Figure 1: Consider ending the caption with "See Fujiwara (2017, Fig.8) for a similar time line but organized per instrument." (this is only a suggestion).*

Done.

*• Section 2.2.1: Consider citing also Manney et al. (JGR, 2007) which provided an excellent overview about the Derived Meteorological Products used here.*

The Derived Meteorological Products described by Manney et al (2007) are not used in this paper.  The PV scaling used here was discussed in that paper, but more thoroughly in the earlier papers that we already cite.

*• Figure 2 and first paragraph of section 3: I do not think that these are really useful (they explain obvious concepts). Consider removing. If you decide to keep, line 23: replace "To demonstrate..." by "To illustrate...".*

The original Figure 2 and corresponding text have been removed considering both your suggestion and that of the other reviewer to do so.

*• P.11, lines 19–21: AIRS was introduced on September 2002 in MERRA and MERRA-2 and on February 2003 in ERA-I and CFSR. The potential importance of this instrument is an interesting point which deserves a few more details. For example, add a reference about its sensitivity to stratospheric temperatures. Similarly interesting comments could be added about GPS-RO which saw assimilation into ERA-I and CFSR from 2001, into MERRA-2 from 2004, into JRA-55 from 2006 and never into MERRA.*

We are now more specific about the introduction of AIRS data in different reanalyses and added a reference (Hoffman and Alexander 2009) that demonstrates AIRS' sensitivity to stratospheric temperature. We believe that ERA-I and CFSR/CFSv2 started assimilating AIRS in 2004, not 2003. MERRA-2 actually began assimilating AIRS in September 2002, not in 2003 as we mistakenly stated.  We added two sentences about GNSS-RO data, including the years when they were assimilated and impact. We cite Fujiwara et al. 2017 for details.

*• P.12, line 30: "...(as was the case in the SH) they remain larger in CFSR..."*

Done

*• P.12, line 35: "...a clear decrease in them is not evident". Please re-phrase.*

This sentence was eliminated in the revised manuscript.

*• Discussion of figure 8 and 9 (p.13 line 26 to p.14 line 13): the standard deviations of the differences are not discussed at all, even though striking differences can be sen with the corresponding Tmin diagnostics in both hemispheres (i.e. compare right columns of fig.5 with fig.8 and fig.6 with fig.9). If you decide to keep showing standard deviations of*

*differences (see first major comment) it would make sense to discuss this.*

We agree. A discussion of standard deviations in A_NAT is included in  both the SH and NH paragraphs. The final paragraph of this subsection notes the overall agreement and discrepancies between the patterns of statistical significance and standard deviations between the two diagnostics and ascribes them to minor differences between reanalyses in the morphology of the fields. The latter point was already made in the original manuscript but only for the SH.

• *P.14 line 29: remove words "... show that the variances of the differences..."*

Done. The sentence now reads: ""The standard deviations tend to increase with height (...)

• *P.15 line 6: see corresponding major comment (2.3)*

As per our discussion of the major comment, the MERRA comparisons have been removed from the paper, so we deleted this text.

• *P.15 line 37: "... (indicating that these reanalyses haveing higher larger SVA than MERRA-2)..."*

This text has been modified due to the switch to using the REM rather than MERRA-2 as a reference.

• *Section 3.3 is very long and tedious to read, probably because it includes the methodology about the diagnostics shown here. Consider moving this methodology to a new subsection in section 2.*

We have moved the paragraphs describing the methodology for these diagnostics to a new subsubsection within section 2.

• *Figures 16–17 and P.19 line 19: It looks like summing the number of days over lower stratospheric levels implies a close dependence on the vertical pressure grid used for this diagnostic. Is there a way to avoid this? In any case the explicit listing of these pressure levels is even more necessary (see major comment 2.4).*

This diagnostic does inherently depend on the resolution and number of pressure levels. There are probably good ways to remove this dependence (e.g., by taking a column average number of days, or selecting the column maximum), but we have removed (what were formerly) Figures 16 and 17 and their discussion to help shorten the paper as requested by reviewer 1, and because the original Figures 18 and 19 provided much the same information in the context of the intercomparisons.

*• Figures 16–19: I understand that plots (b) and (d) simply show the same sensitivity range as already shown by the bars in plots (a) and (d) but zoomed and centered on zero? If this is wrong, the captions and text require clarification. If this is right, the usefulness of plots (b) and (d) is not clear since they could be removed while not changing the discussion of the figures (also because the figures show sensitivities which do not depend much on the reanalyses nor on the year).*

Your first interpretation is correct; that is, panels b and d show the same sensitivity ranges, but centered on zero. We include them because the human eye has trouble properly assessing the lengths of bars/lines when they are in different contexts, such as being centered at different heights as in panels a and d (see also, e.g., the Ponzo illusion and/or the Müller-Lyer illusion). We have kept these panels because we do discuss them specifically (however, we have made these references more apparent), and we think it is important to show how the sensitivities compare among the reanalyses.

*• P. 17 line 19: After 1999, it looks like fig. 17c has quasi-biennal periodicity. Could there be a link between this diagnostic and the QBO?*

We do plan to look more in depth at potential links between the QBO and polar processing diagnostics with some other S-RIP collaborators, but for now, this is beyond the scope of the paper.

*• P.17 line 21: Why is the important diagnostic VPSC/Vvort and not VPSC itself? If possible, explain this in one additional sentence.*

We have reworded this and added a phrase to note that this scaling is used to get a diagnostic that is independent of the (considerable) interannual and interhemispheric variations in vortex size.

*• P.17 lines 27–28: "...with a range among the reanalyses of 0.98 – 1.30 km..."*

This is not accurate because the altitude differentials are independent of the reanalyses. The Knox approximation is a simple approximation to go from potential temperature to altitude. Since we are vertically integrating areas to get volumes, we need the altitude widths of the potential temperature levels (i.e., the altitude differentials, or thinking in terms of integrals, the "dz" values). We have clarified the text to make this more explicit that we are referring to the minimum and maximum altitude differentials.

*• P.17 lines 28–29: I expect that the winter mean is applied at the end, i.e. you discuss winter means of daily fractions rather than the fractions of winter mean volumes? Please clarify, if possible in the caption of fig. 18 as well.*

Yes, this is correct. We have clarified the text to make this explicit.

• P.18 lines 3–4: this last sentence is not useful (see also comment above about plots (b) and (d) not useful in these figures).

We have reworded this sentence to make it clearer that we consider this worth saying as it is evidence against persistent biases in horizontal temperature gradients.  Also see our response to your comment on Figure 16-19 above.

• P.18 lines 11–12: "...with differences betweeen reanalyses indicating some differences in horizontal temperature gradients (especially in, e.g., 2011 and 2014).". Can this be seen directly on fig. 19 or are you commenting figures which are not shown in the manuscript? Please clarify.

Persistent biases in the temperature gradients would result in different sensitivity to the changes in PSC threshold values used.  In addressing the previous comment, we have revised the text to make this more explicit.

• Figures 20–21: please align the titles of the figures with the vocabulary used in the text (i.e. vortex decay dates - not vortex breakup dates). Since the ranges are the same for both figures, it is possible to clarify the caption: "...differences that greatly exceed the range by more than 7 days larger than 21 days are marked with a white X."

The figure titles and captions have been modified along the lines requested.

• P.18 line 35: "For the SH, Figure 20a shows that..."

Done.

• P.19 line 8: It is possible to get away from the figure and closer to its meaning. For example: "Fig.20 also shows that on a few years (such as 2002 and 2009) the vortex decayed at a much later date in MERRA, JRA-55 and CFSR than in MERRA-2.

This text was changed in switching from looking at differences from MERRA-2 to differences from the REM.  We have tried to word all this discussion in terms of earlier/later vortex decay dates rather than positive/negative differences.

• P.20 line 7: It is not fair to compare agreement generally within about 1K after 1998 with disagreements up to about 6K before.

The numbers have been altered by using the REM, but we have reworded the corresponding sentence to specify the range of differences in a consistent way in each case.

• P.22 line 34: "...and after rigorous assessment of the relationships of temperature changes

*to observations assimilated (which, to our knowledge has not been done)." It could be that this has not yet been done systematically due to the huge diversity of assimilated observations. Yet Simmons et al. (QJRMS, 2014, doi:10.1002/qj.2317) already provided a remarkable first step in this direction.*

Simmons et al. did, indeed, provide a fairly detailed analysis of responses to inputs for ERA-Interim, and compared the results for long-term variations with that in ERA-40 (now obsolete), MERRA (now becoming obsolete), and JRA-55.  They did not do such a detailed analysis of the other reanalyses, nor include MERRA-2 (not yet available at that time) or CFSR/CFSv2.  We had overlooked mentioning this study in the introduction when we discussed previous intercomprisons, an omission that we have remedied.  We have qualified the statement here to read "..has not been done for most of the reanalyses considered here."

*• P.23, last sentence: "...the comparison of reanalyses is a powerful tool for assessing robustness and uncertainty in these diagnostics." Yes but this is still an incomplete tool because the reanalyses often use similar parametrizations and assimilate very similar observational datasets.*

This point is, indeed, made in several places in the text of the paper.  We feel this statement (which does not imply that it is the only powerful tool or that it is a perfect tool) is appropriate for closing the text of the paper.

---

## Author Response (AR2)

**Response to Editor's comments:**

*The authors should address the comments from Referee #2, as well as the following points:*

See responses to Referee #2.

*P. 2, L. 5: Perhaps the authors should indicate in the abstract what is the "profound effect"?*

We have changed the wording to indicate that there is a profound improvement in the agreement.

*P. 4, L. 1: "great hesitancy" sounds a bit odd to me. Could the authors use a different expression, perhaps, "great care"?*

We have changed this to "great caution".

*P. 16, L. 12, 14, 19: Not statistically significant at which level? Elsewhere, if possible, quantify these statements on statistical significance.*

As noted in the methods section, we assess the statistical significance at the 99% confidence level. We have added an explicit note to this effect in the methods section and a reminder to the reader at the first use of "statistically significant" in the temperature and vortex subsections of the results (including on page 16).

*P. 22, L. 18: Do you need to use "etc"?*

We have reworded this to say "factors such as…" and dropped "etc".

*P. 22, L. 30: Perhaps the use of "vast" is a bit hyperbolic. Could the authors use a different word?*

We have changed "vast" to "very large".

*P. 24, L. 9: "Nice" is subjective. Could the authors use a different word? If not, perhaps they could define what they mean by "nice".*

We have added a definition of what we mean by "nice".

*P. 34, Fig. 2 caption: If the authors do not include details of the colour styles in the caption, but instead use a legend embedded in the figure, I suggest they consider including in the caption a reference to the legend. Same for other similar figures.*

We have added a note giving the colors used for each reanalysis to the Figure 2 caption.   The other figures that use these colors already had such text in the captions.

*P. 35, Fig. 3 caption: I suggest the authors consider including information on the end-points of the colour scale. Same for other similar figures.*

We have added information on the endpoints of the color range to the Figure 3, 6, 9,and 12 captions.

*P. 49, Fig. 17 (and 18): I find it difficult to see the white X. Could the authors address this?*

When we switched to looking at differences from the REM, the values turned out so that there are no whites Xs in Figure 17 (hence the difficulty in seeing them).  There are several in Figure 18, which we believe are easily visible in the high-resolution figures we are providing.  We have moved the description of the white Xs from the Figure 17 to the Figure 18 caption.

**Response to Referee #2's Comments:**

*This is my second review of this manuscript. I thank the authors for their careful consideration of the comments of both reviewers, to my reading the manuscript is clearer and more digestible than the first version. I am happy to recommend it for publication essentially as is - I have one overall thought which the authors may which to consider and two very minor textual points.*

We thank the referee for their helpful reviews.

*The authors at various points discuss the value of reanalysis intercomparisons, generally in a very positive light. If I understand correctly, the basic argument is that, for highly derived quanities of relevance to polar processing, or for quantities that otherwise remain underconstrained by observations, this kind of reanalysis intercomparison serves to quantify the state of our knowledge in a way that would be otherwise very difficult to do. If multiple, 'independently' produced reanalyses agree, this gives confidence that the available observations constrain the quantity in question. I agree that this is a powerful methodology, but the point that Simon brought up is an important considerationn; there are plausible reasons for reanalyses to be wrong in the same way. I bring this up only suggest that it may be worth being a bit more explicit somewhere about the underlying philosophy. Perhaps it is all in the text as it stands but I think my comment in the previous review about reanalyses being the 'best' tool available was possibly a result of me taking the comment more broadly than it was intended.*

This is indeed an important point to keep in mind when comparing the reanalyses. The revised paper already discussed this in the "Recommendations" in the conclusions section (around lines 10-16 of page 23 of that version). We have also added notes to this effect in the introduction (page 4, line 1) and data (page 7, lines 24--26) in the final version of the manuscript.

*p4, l8 'do/did not' is confusing here - is the intent to say some limb-sounding*
*satellites did not retrieve temperaures low enough and those that do typically*
*have incomplete coverage?*

Yes, the referee is correct about our intended meaning. We have re-worded this sentence to make our meaning more explicit.

*p8 l19: perhaps more precise to say they are simply given in table A2?*

We now say "... shown in Appendix A (Table A2)."

[revised manuscript text omitted]

(2017), as well as similarities in data inputs that could results in biases in all of the reanalyses, argue for great  caution in using temperatures and other fields from individual reanalyses for diagnosing long-term changes and trends.

The ability to compare polar processing diagnostics with observations is very limited for several reasons. Somewhat paradoxically, the vast improvements in DAS usage of available observations have resulted in there being very few truly independent temperature datasets. Furthermore, many of the datasets that are available, even those ingested into the DAS, generally suffer from very limited spatial and/or temporal coverage (e.g., balloon-borne and lidar measurements) and/or issues with resolution, precision, and length of data records (e.g., limb-sounding research satellites). For example, many  previous and current limb-sounding satellites had/ have very coarse vertical resolution, and did not/do not retrieve temperatures to low enough altitudes to fully cover the lower stratosphere; those that do cover the lower stratosphere 
[revised manuscript text omitted]

[Figure]

**Figure 4.** SH winter season (MJJASO) (a, c, e, g) averages and (b, d, f, h) standard deviations of minimum temperature differences for each reanalysis from the reanalysis ensemble mean (REM) as a function of year and pressure for the 1979 through 2015 winters, concatenated from the individual years into pixel plots as described in the text. Columns of grey pixels indicate years with no data. Pixels with x symbols inside indicate years and levels where the differences from the REM are insignificant according to our bootstrapping analysis (see section 2.2.4). Blues in the average difference panels show negative values (reanalysis less than the REM) and reds positive values (reanalysis greater than the REM); in the standard deviation panels, yellows/deep blues represent low/high standard deviations of the reanalysis differences, respectively.

[Figure]

**Figure 5.** As in Figure 4 but for the NH winter seasons (DJFM) for 1979/1980 through 2015/2016. Note that different color ranges are used for the NH shown here than in Figure 4 for the SH.

[Figure]

**Figure 6.** As in Figure 3, but for area with temperatures below the NAT PSC threshold. The color range in panels a and b goes from 0 to 12% of a hemisphere (blues to reds), with all values above 12% shown in the deepest red.

[Figure]

**Figure 7.** As in Figure 4 but for area with temperatures below the NAT PSC threshold in the SH.

[Figure]

**Figure 8.** As in Figure 7, but for the NH. See text for explanation of date ranges used for the calculations.

[Figure]

**Figure 9.** As in Figure 3, but for maximum sPV gradients. The color range in panels a and b is from 0 to 20 PVGU (blues to reds), with all values over 20 shown in the deepest red.

[Figure]

**Figure 10.** As in Figure 4, but for maximum sPV gradients.

[Figure]

**Figure 11.** As in Figure 10, but for the NH.

[Figure]

**Figure 12.** As in Figure 9, but for sunlit vortex area. The color range in panels a and b is from 0 to 13% of a hemisphere (blues to reds), with all values over 13% shown in the deepest red.

[Figure]

**Figure 13.** As in Figure 4, but for SH sunlit vortex area.

[Figure]

**Figure 14.** As in Figure 13, but for the NH.

[Figure]

**Figure 15.** Winter means of the fraction of vortex volume between the 390 and 580 K isentropic surfaces with temperatures below $T_{ice}$ in the SH (a and c), and range of values obtained for the $\pm 1$ K sensitivity tests (b and d). The colored bars show the range of values obtained from the tests of sensitivity to the PSC threshold temperature used (see Section 2.2.2), while the black dots show the value for the "central" threshold temperature. The winter mean is calculated over the full MJJASO period. For each year, the reanalyses are ordered from smallest central value on the left to largest central value on the right; this order is also given as a text string at the top of the column for each year. The numbers at the bottom of each year's column indicate the difference in winter mean fraction between the largest and smallest central values for the winter season (i.e., between the rightmost and leftmost black dots). In the range panels (b and d), the range about the central value (black dots in a and c) is shown for each reanalysis. Green, blue, purple, and red indicate CFSR/CFSv2, ERA-I, JRA-55, and MERRA-2, respectively.

**NH, DJFM Mean $V_{NAT}$ / $V_{Vort}$**

[Figure]

**Figure 16.** As in Figure 15 but for the NH and temperatures below $T_{NAT}$

[Figure]

**Figure 17.** Pixel plots of (a) vortex decay dates (see text for the definition) based on the reanalysis ensemble mean (REM) of vortex area, and (b through e) the difference between the vortex decay dates in each of the reanalyses from the REM (as reanalysis − REM).

[Figure]

**Figure 18.** As in Figure 17 but for the NH. The color bar ranges are restricted to distinguish differences of a few days; differences whose magnitude greatly exceeds the range (by more than 7 days, thus differences with magnitude greater than 21 days) are marked with a white X.

**Table A1.** $HNO_3$ and $H_2O$ Values for PSC Threshold Calculation. Pressure values are rounded to nearest integer.

| pressure (hPa) | $HNO_3$ (ppbv) | $H_2O$ (ppmv) |
|---|---|---|
| 147 | 3.0 | 4.0 |
| 100 | 3.7 | 4.0 |
| 68 | 6.1 | 4.8 |
| 46 | 9.8 | 5.0 |
| 32 | 12.0 | 5.5 |
| 22 | 11.8 | 5.8 |
| 15 | 10.0 | 5.9 |
| 10 | 6.8 | 6.0 |

**Table A2.** Pressure levels, approximately corresponding potential temperature levels, and the NAT and ice PSC thresholds calculated from Table A1. Pressure values are rounded to the nearest integer; NAT and ice PSC thresholds are rounded to the nearest tenth of a Kelvin.

| pressure (hPa) | potential temperature (K) | NAT PSC threshold (K) | ice PSC threshold (K) |
|---|---|---|---|
| 121 | 390 | 198.2 | 192.4 |
| 100 | 410 | 197.3 | 191.2 |
| 83 | 430 | 197.1 | 190.7 |
| 68 | 460 | 196.6 | 190.0 |
| 56 | 490 | 196.0 | 189.0 |
| 46 | 520 | 195.3 | 188.0 |
| 38 | 550 | 194.6 | 187.2 |
| 32 | 580 | 193.9 | 186.4 |
| 26 | 620 | 192.9 | 185.5 |
| 22 | 660 | 192.0 | 184.5 |
| 18 | 700 | 190.9 | 183.5 |
| 15 | 750 | 189.8 | 182.5 |
| 12 | 800 | 188.6 | 181.5 |
| 10 | 850 | 187.4 | 180.5 |